# TIPS: Text-Image Pretraining with Spatial awareness

**Kevis-Kokitsi Maninis**[*]  **Kaifeng Chen**[*]  **Soham Ghosh**[*†]  **Arjun Karpur**[*]
**Koert Chen**  **Ye Xia**  **Bingyi Cao**  **Daniel Salz**  **Guangxing Han**  **Jan Dlabal**
**Dan Gnanapragasam**  **Mojtaba Seyedhosseini**  **Howard Zhou**  **André Araujo**

Google DeepMind

## ABSTRACT

While image-text representation learning has become very popular in recent years, existing models tend to lack spatial awareness and have limited direct applicability for dense understanding tasks. For this reason, self-supervised image-only pretraining is still the go-to method for many dense vision applications (e.g. depth estimation, semantic segmentation), despite the lack of explicit supervisory signals. In this paper, we close this gap between image-text and self-supervised learning, by proposing a novel general-purpose image-text model, which can be effectively used off the shelf for dense and global vision tasks. Our method, which we refer to as Text-Image Pretraining with Spatial awareness (TIPS), leverages two simple and effective insights. First, on textual supervision: we reveal that replacing noisy web image captions by synthetically generated textual descriptions boosts dense understanding performance significantly, due to a much richer signal for learning spatially aware representations. We propose an adapted training method that combines noisy and synthetic captions, resulting in improvements across both dense and global understanding tasks. Second, on the learning technique: we propose to combine contrastive image-text learning with self-supervised masked image modeling, to encourage spatial coherence, unlocking substantial enhancements for downstream applications. Building on these two ideas, we scale our model using the transformer architecture, trained on a curated set of public images. Our experiments are conducted on 8 tasks involving 16 datasets in total, demonstrating strong off-the-shelf performance on both dense and global understanding, for several image-only and image-text tasks. Code and models are released at https://github.com/google-deepmind/tips.

## 1 INTRODUCTION

The quest for effective image representations has permeated much of the research work in computer vision over the past two decades: starting with hand-crafted techniques such as SIFT (Lowe, 2004) and HOG (Dalal & Triggs, 2005), then moving into the deep learning era with supervised (Krizhevsky et al., 2012; He et al., 2016; Dosovitskiy et al., 2021), weakly-supervised (Radford et al., 2021; Mahajan et al., 2018) or self-supervised (Chen et al., 2020; Caron et al., 2021; He et al., 2020) techniques. Most computer vision tasks critically depend on capable image encodings, and for this reason the holy grail in image representation learning is a generic model that can be used off the shelf for a variety of downstream tasks.

One of the most promising directions for such representation learning research is on leveraging noisy textual supervision, which is abundant on the web, as introduced by CLIP (Radford et al., 2021) and ALIGN (Jia et al., 2021). Recent methods (Zhai et al., 2023; Sun et al., 2023) have also made progress on image-text learning. However, these techniques have for the most part not been successful at dense image prediction tasks, such as depth estimation or semantic segmentation. On the other hand,

---

[*]Authors contributed equally.
[†]Soham Ghosh is now with Mistral AI.
Correspondence: {kmaninis,francischen,andrearaujo}@google.com

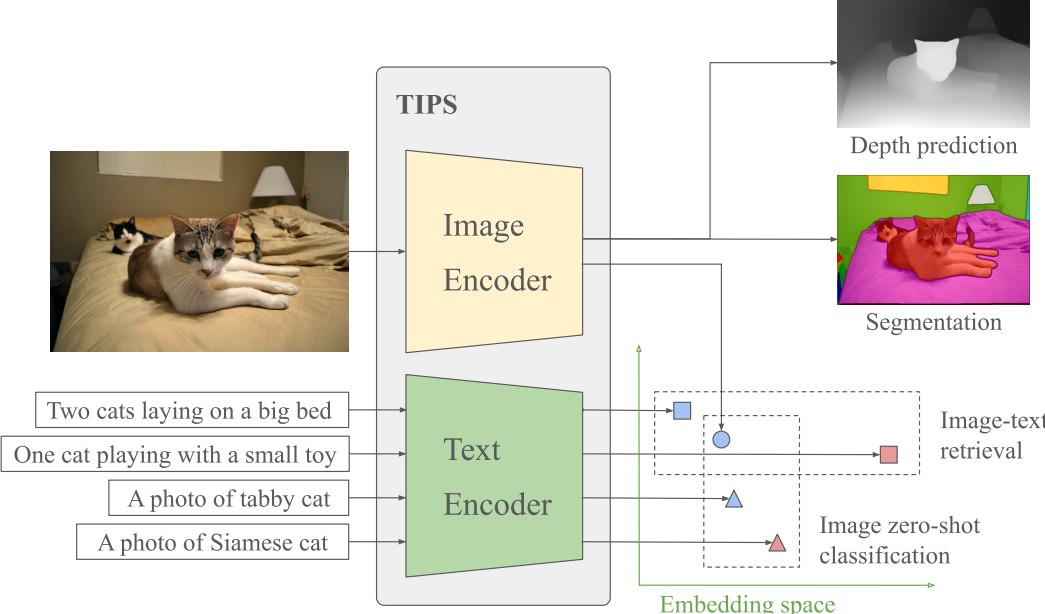

Figure 1: We introduce **TIPS: Text-Image Pretraining with Spatial awareness**. TIPS is a general-purpose image-text encoder model, which can be effectively used for dense and global understanding, in vision-only or vision-language tasks.

self-supervised learning techniques (Caron et al., 2021; Zhou et al., 2022), although lacking semantic signals to guide the training, can enforce feature consistency between distorted images or spatially adjacent patches, and result in effective pretraining techniques for dense image understanding (Oquab et al., 2024). These methods, however, do not use any text for training, and are by nature limited to vision-only tasks. In this work, we build on top of both the image-text and self-supervised learning paradigms, to develop general-purpose image representations which can be used for a variety of image-only and image-text tasks. We build strong representations for dense prediction tasks, such as segmentation and depth estimation, and global prediction tasks that reason about the image as a whole, such as image classification and image-text retrieval. We illustrate our method in Fig. 1.

We address the limitations of image-text learning, which hinder its applicability to dense spatial understanding tasks, with two simple and effective ideas that improve weak supervision and incentivize image features to become spatially coherent: **(1)** Enhancing the textual labels via automated generation of image captions, leveraging a recent multimodal generative model (Beyer et al., 2024). Such synthetic image captions tend to describe the visual contents more comprehensively than the noisy captions mined from the web, by capturing all the objects in the scene and their spatial relationships, being a rich supervision signal for dense understanding. However, at the same time, noisy web captions often contain fine-grained details which can be helpful for global understanding tasks (as discussed in Fig. 3). Thus, we devise an effective method to train our model with both noisy and synthetic captions, with separate image-text contrastive losses for them, to achieve strong performance on both dense and global tasks. **(2)** Encouraging the learned image features to be spatially coherent, inspired by lessons from the self-supervised literature. We incorporate self-distillation and masked image modeling (MIM) into the image-text learning framework, with carefully designed adaptations, resulting in substantial improvements for many applications, especially the dense ones.

Finally, we build on these ideas to scale a Vision Transformer (Dosovitskiy et al., 2021) with text alignment on a training dataset of 117M public images, leveraging their noisy web captions and synthetically-generated ones. Our method showcases spatial understanding and textual alignment *in the same model*, essentially combining the strengths of the image-text and self-supervised literature. We refer to our method as **T**ext-**I**mage **P**retraining with **S**patial awareness (TIPS), and thoroughly evaluate it across many downstream tasks. Specifically, we demonstrate that TIPS achieves strong and competitive performance off-the-shelf across 8 computer vision tasks involving 16 datasets in total, comprising image-only or image-text evaluations, for dense or image-level predictions. We hope that our findings will inspire the community towards the development of next-generation image representations, to enable multimodal and spatially grounded applications.

## 2 RELATED WORK

**General-purpose image representation models** have been proposed for computer vision tasks, generally leveraging self-supervised or weakly-supervised learning. Recent self-supervised techniques include DINO (Caron et al., 2021), MAE (He et al., 2022), iBOT (Zhou et al., 2022), I-JEPA (Assran et al., 2023) and the scaled-up DINOv2 (Oquab et al., 2024), which employs a large curated dataset. Our work differs from self-supervised approaches by learning with readily-available and public textual captions, which makes the model more capable as it can handle language inputs. Weakly-supervised learning of image representations generally leverages noisy textual captions, with early examples coming from Joulin et al. (2016); Mahajan et al. (2018). Modern approaches include CLIP (Radford et al., 2021), ALIGN (Jia et al., 2021), COCA (Yu et al., 2022), OpenCLIP (Cherti et al., 2023), SigLIP (Zhai et al., 2023), Florence (Yuan et al., 2021), InternVL (Chen et al., 2024) and the EVA series (Fang et al., 2023; 2024; Sun et al., 2023). Different from these image-text techniques, we design TIPS to offer frozen pretrained image features that are directly useful to a broad range of downstream vision tasks, without model fine-tuning. Generally, existing image-text models do not focus much on downstream dense prediction tasks with frozen features. Additionally, our technique differs from these by enhancing the core contrastive training common to all of these methods to obtain spatially-coherent representations, via improved captions and loss functions.

**Image-text learning for dense understanding tasks.** While the above-mentioned existing image-text learning approaches lead to powerful representations, they have not demonstrated clear benefits for dense image prediction tasks. As a consequence, today the models learned with self-supervised techniques are preferred for these cases: for example, the recent DepthAnything (Yang et al., 2024) model is built on top of the self-supervised DINOv2 features, even though weakly-supervised image backbones are widely available. Similarly, Tong et al. (2024) have demonstrated shortcomings of CLIP-style models and incorporated DINOv2 to enhance the visual grounding of multimodal models. However, recent work has adapted image-text learning for dense prediction, e.g. for open-vocabulary detection (Kim et al., 2023; Minderer et al., 2022; Rao et al., 2022) and segmentation (Mukhoti et al., 2023; Wu et al., 2024; Wysoczanska et al., 2024). SLIP (Mu et al., 2021) combines an adapted SimCLR (Chen et al., 2020) self-supervised objective with CLIP for classification tasks. Closer to our work, MaskCLIP (Dong et al., 2023) leverages masked image modeling with contrastive learning, and the very recent SILC method (Naeem et al., 2024) combines contrastive image-text training with self-distillation. Different from our goals, most of these methods are specifically tailored towards improving vision-language tasks that require spatial understanding and do not aim to learn a general-purpose vision encoder. Besides, most of these do not incorporate dense objectives during pretraining, and may require additional fine-tuning stages or dense supervision, which can be costly. MaskCLIP and SILC propose to incorporate self-supervised losses into image-text training, but only employ either masked image modeling or self-distillation, respectively; in contrast, our TIPS technique goes beyond to combine both, which we show to boost performance significantly for dense image prediction. FLIP (Li et al., 2023) proposed to combine contrastive learning with masking, but without any reconstruction loss, aiming only at efficient language-image training. Altogether, our approach demonstrates for the first time a way to learn image-text models whose vision representations rival those of self-supervised approaches in dense image understanding tasks.

**Synthetic data for image representation learning.** One of our contributions is to show the power of synthetic textual captions for image representation learning, in particular for dense prediction. Previous work has explored the training of visual representation models with synthetic data, especially with synthetic images (Ren & Lee, 2018; Tian et al., 2023; Sariyildiz et al., 2023), which may be generated based on LLM captions (Tian et al., 2024; Hammoud et al., 2024). CapsFusion (Yu et al., 2024) leverages synthetic captions to improve large multimodal models for generative text applications. More similar to our work, VeCLIP (Lai et al., 2024) and LaCLIP (Fan et al., 2023) generate synthetic captions for contrastive image-text training. However, their methods are only applied to global image understanding tasks such as image retrieval and classification, and there is no consideration related to dense prediction. In contrast, we reveal for the first time the power of synthetic captions to improve spatial understanding in image-text models. Additionally, we propose a new learning method to combine noisy web captions with synthetic descriptions, by introducing an additional vision transformer class token to better leverage synthetic descriptions, boosting performance in many tasks. Our technique goes beyond the sampling or multi-text caption combination strategies proposed in LaCLIP, enabling more flexible learning with two image-level tokens which focus on different characteristics.

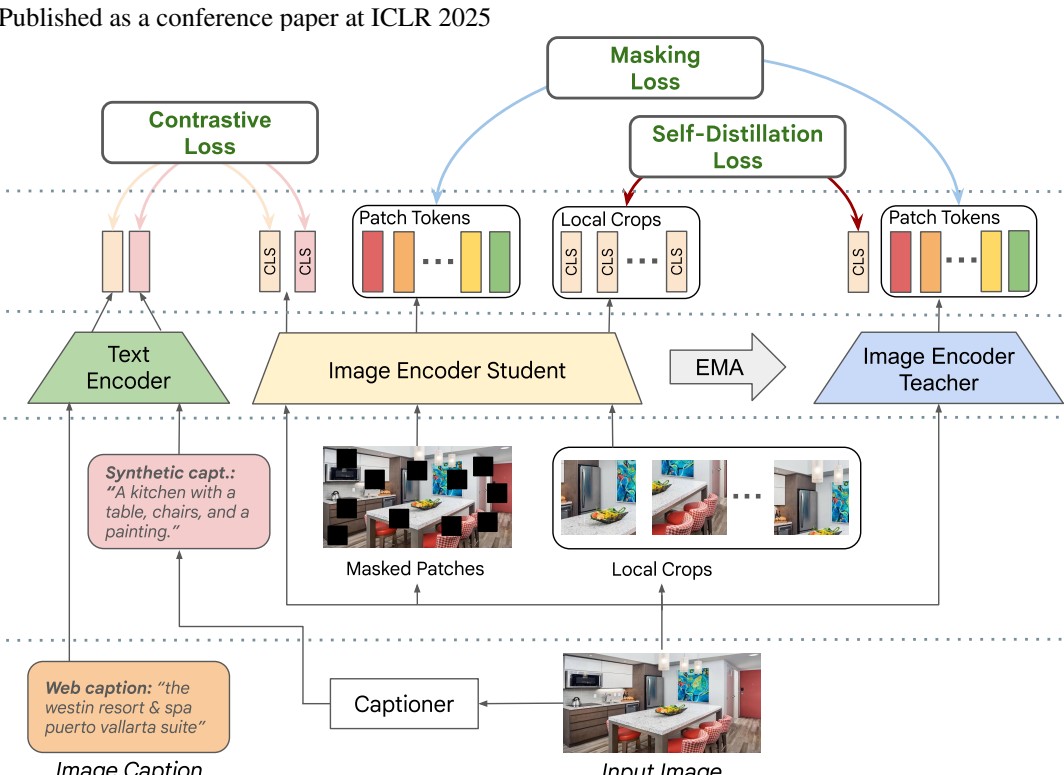

Figure 2: **Block diagram of TIPS.** From bottom to top: given an input image, we produce masked and cropped augmentations, along with synthetic descriptive captions from a captioner model. They are fed into the text and image encoders, along with the noisy web caption, and the output tokens are used in the losses. The contrastive loss makes use of the two captions, aligning them with two [CLS] tokens obtained from the image encoder. TIPS also employs self-distillation applied to the local crops and a masked image modeling loss applied to dense patch tokens, which encourage spatially-aware and discriminative image representations.

## 3 TIPS

Our goal is to create a general-purpose image representation model, with text alignment, which can be used off-the-shelf for dense and global vision tasks. While image-text contrastive techniques (Radford et al., 2021; Jia et al., 2021) can effectively model global image information, they tend to underperform for dense understanding tasks, where self-supervised models are the method of choice today (Oquab et al., 2024). To bridge this gap, we propose Text-Image Pretraining with Spatial awareness (TIPS), illustrated in Fig. 2, which leverages enhanced weak supervision via synthetic image captions, as well as self-supervised masked modeling, improving image feature quality significantly, for both dense and global understanding.

**Problem setup.** Given a collection of image-text pairs $\{(I_k, T_k)\}$, where $T_k$ is a noisy textual caption for image $I_k$, we aim to learn a model which encodes images into dense and global embeddings that are useful to a variety of multimodal tasks. More concretely, we set out to train the function $f$, mapping image $I$ to a set of image embeddings $\{\mathbf{e}^g, \mathbf{e}_1, \mathbf{e}_2, \ldots, \mathbf{e}_N\}$, where $\mathbf{e}^g$ is the global embedding representation of the entire image and $\{\mathbf{e}_n\}_{n=1}^N$ are patch embeddings corresponding to different image regions. The text associated with the images can be leveraged to train a semantically meaningful joint embedding space, leading to useful image features. We build on top of the standard CLIP method (Radford et al., 2021), which learns a text encoder $g$, mapping $T$ to its embedding $\mathbf{e}^t$, by pushing $\mathbf{e}^g$ and $\mathbf{e}^t$ close for corresponding images and captions, and far otherwise. CLIP uses a cross-entropy loss with softmax normalization of cosine similarities, referred to as InfoNCE (van den Oord et al., 2018), which we denote $\mathcal{L}_{CLIP}$. In this work, we model $f$ as a Vision Transformer (ViT) (Dosovitskiy et al., 2021) and obtain the image embeddings from the final layer's feature map, with $\mathbf{e}^g$ corresponding to the [CLS] token. The function $g$ is modeled as a standard transformer (Vaswani et al., 2017).

### 3.1 ENHANCING WEAK SUPERVISION WITH SYNTHETIC IMAGE CAPTIONS

A limitation of standard image-text learning using large-scale web data is the quality of the captions, which are noisy and may not accurately describe images. An example is shown in Fig. 3 (top),

where the words "for sale dealership \$30k" are not describing the image contents. While this may hinder model learning, the caption still captures the main object ('*2007 Cadillac Escalade*').

However, a deeper issue we commonly observe is that these captions often only mention salient objects, without describing their arrangement in the scene. In other words, the captions usually serve as noisy image-level supervision and generally tend to be of limited use for learning spatially-aware representations. This motivates us to investigate automated generation of synthetic captions, which could serve as useful pre-training weak supervision for dense tasks.

We employ off-the-shelf, publicly available models which can caption images effectively: given the image $I$, we generate text $\hat{T}$. In particular, we leverage captioning models which tend to generate accurate and high-level image descriptions – an example is given in Fig. 3 (bottom). Note the use of the preposition "in front of", which indicates the spatial arrangement of the scene, the description of the background ("building") and the color of the object ("black"), all combined providing rich signals for dense image representation learning.

Noisy web caption: *"2007 Cadillac Escalade for sale dealership \$30k"*

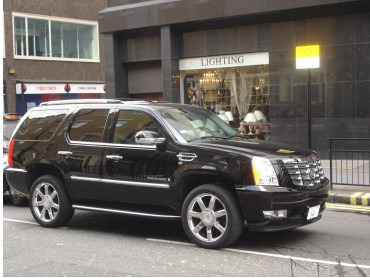

Synthetic caption: *"A black SUV parked in front of a building."*

Figure 3: Example web image (CC BY-SA 2.0) with noisy caption (top) and synthetic caption by PaliGemma (Beyer et al., 2024) (bottom).

However, a drawback of the synthetically generated captions is their lack of detailed object semantics. Referring again to Fig. 3, the synthetically-generated caption misses information of the specific car model (*2007 Cadillac Escalade*), which can be helpful to learn discriminative representations. For this reason, we propose to combine the original $T$ and synthetic $\hat{T}$ captions, to aim for globally discriminative and spatially-aware image features.

**Dual image-text embedding.** We strive to leverage relevant information from both captions, and thus propose to modify the vision transformer $f$ to learn from them, in an approach we call "dual embedding". We insert an additional [CLS] token in the model, to be used for learning with the synthetic caption, obtaining an additional global embedding $\hat{\mathbf{e}}^g$. At training time, we feed both $T$ and $\hat{T}$ into the text encoder, to obtain their text embeddings $\mathbf{e}^t$ and $\hat{\mathbf{e}}^t$. In addition to the $\mathcal{L}_{CLIP}$ loss between $\mathbf{e}^g$ and $\mathbf{e}^t$, we compute $\hat{\mathcal{L}}_{CLIP}$ between $\hat{\mathbf{e}}^g$ and $\hat{\mathbf{e}}^t$. This introduces flexibility in the model to learn an object-centric image embedding in $\mathbf{e}^g$, and a more spatially-aware image embedding in $\hat{\mathbf{e}}^g$. Both back-propagate into the dense feature maps to learn improved patch embeddings $\{\mathbf{e}_n\}_{n=1}^N$. At inference time, the model can have access to both types of global image embeddings, and the one to use may be decided based on the downstream task: generally spatially aware tasks will use $\hat{\mathbf{e}}^g$ while object-centric ones will employ $\mathbf{e}^g$.

## 3.2 Integrating Self-Distillation and Masking to Boost Image Features

In addition to improving training data quality and learning with different types of textual supervision, we propose to incentivize the model to learn spatially-aware representations via dedicated loss functions. We are inspired by recent self-supervised learning techniques, which produce features suitable to dense downstream tasks (Caron et al., 2021; Zhou et al., 2022; Oquab et al., 2024). We incorporate self-distillation and masking losses in our training setup, adapting them to work in a weakly-supervised image-text learning framework. Building on top of CLIP, we introduce a teacher ViT model, $f_t$, to help guide the training process, which processes the full image $I$. The teacher's weights are updated by Exponential Moving Average (EMA) of the main (student) ViT, $f_s = f$, as per (He et al., 2020). Two additional loss terms are introduced, as described next.

**Self-distillation loss.** We create $M$ local crops from the input image $I$, which are processed by $f_s$, to obtain $M$ local crop embeddings via their [CLS] tokens, $\{\mathbf{e}^{g,m}\}_{m=1}^M$. During training, we enforce these embeddings to match predictions of the teacher's [CLS] token, $\mathbf{e}^{g,t}$, which is obtained from a forward pass of $I$ through $f_t$. This incentivizes the model to learn representations which are consistent across the local crops and the original (global) image. The embeddings are used to compute prototype scores using an MLP-based projection head, on top of which softmax normalization and cross-entropy loss are applied:

$$\mathcal{L}_{distill} = -\sum_b \sum_m \text{softmax}((\mathbf{p}_b^t - \mathbf{c})/\tau_t) \log(\text{softmax}(\mathbf{p}_b^m/\tau_s)) \tag{1}$$

where $b$ iterates over the images in the batch. $\mathbf{p}^t = \text{P}_t(\mathbf{e}^{g,t})$ and $\mathbf{p}^m = \text{P}_s(\mathbf{e}^{g,m})$ correspond to the teacher's and student's prototype scores respectively, which are computed with the teacher and student projections, $P_t$ and $P_s$, where $P_t$ is updated with an EMA of $P_s$. $\tau_t$ and $\tau_s$ correspond to the teacher's and student's temperatures, used for sharpening the scores, and $\mathbf{c}$ to a centering variable which is updated with an EMA of $\mathbf{p}_b^t$, to encourage a uniform distribution.

**Masking loss.** We introduce a masked image modeling loss in order to encourage the learned patch embeddings to understand their spatial surroundings. The high-level idea is to have the visible patch representations recover the semantics of the masked patches. More concretely, we feed a masked version of $I$ through $f_s$, where the masked patches are replaced by mask tokens, $\{\mathbf{m}_n\}_n$. The encoded mask tokens, $\{\mathbf{e}_n^m\}_n$, are then projected to prototype scores and compared to the teacher's corresponding unmasked tokens, $\{\mathbf{e}_n^t\}_n$, similarly to Eq. 1:

$$\mathcal{L}_{mask} = -\sum_b \sum_n \text{softmax}((\mathbf{p}_{b,n}^t - \mathbf{c}')/\tau_t') \log(\text{softmax}(\mathbf{p}_{b,n}^m/\tau_s')) \tag{2}$$

where, again, $b$ iterates over the batch. $\mathbf{p}_n^t = \text{P}_t'(\mathbf{e}_n^t)$ and $\mathbf{p}_n^m = \text{P}_s'(\mathbf{e}_n^m)$ correspond respectively to the teacher's and student's prototype scores for patch $n$, which are computed with the teacher and student projections, $P_t'$ and $P_s'$, where $P_t'$ is updated with an EMA of $P_s'$. Similarly as before, $\tau_t'$ and $\tau_s'$ correspond to the teacher's and student's temperatures and $\mathbf{c}'$ to the centering variable.

The total loss for our method is then: $\mathcal{L}_{total} = \frac{1}{2}(\mathcal{L}_{CLIP} + \hat{\mathcal{L}}_{CLIP}) + \alpha\mathcal{L}_{distill} + \beta\mathcal{L}_{mask}$.

**Discussion.** Our method builds on ideas from the weakly and self-supervised literature, and to the best of our knowledge is the first to demonstrate that simultaneously combining contrastive image-text learning with both self-distillation and masked image modeling leads to improvements across many tasks, indicating positive synergies between these objectives. The closest existing techniques are MaskCLIP (Dong et al., 2023) and SILC (Naeem et al., 2024), which combined CLIP with either masked image modeling or self-distillation. As we show in experimental ablations, though, combining the masked image loss with self-distillation substantially improves performance across dense tasks, being critical for downstream applications. We also note some key differences compared to previous methods. Given that we use a CLIP loss, the self-supervised components can be simplified, compared to the original formulations in DINO (Caron et al., 2021) and iBOT (Zhou et al., 2022). A major difference is that we use a single global "crop", instead of two in DINO, iBOT and SILC, substantially increasing throughput by 25%. In contrast to many self-supervised methods, we use comparatively simple data augmentations: the local crop is just a random crop of the original image, and the global crop is a larger random crop with horizontal flips. This is similar to Assran et al. (2023); Moutakanni et al. (2024) who argue that complex augmentations may not be necessary for representation learning. Finally, our masking approach is simply random, in contrast to blockwise in iBOT.

## 3.3 SCALING TIPS

We aim at creating a highly-capable and general-purpose model, and for this reason it is critical to scale it to a large model architecture and training dataset, aiming at enhanced image representations.

**Model.** The ViT architecture has been shown to scale well to billion-sized models in a variety of tasks (Zhai et al., 2022; Oquab et al., 2024; Chen et al., 2024). We scale our TIPS model to the ViT-g architecture, with patch size 14, and use the SwiGLU (Shazeer, 2020) feed-forward network variant. Similar to Oquab et al. (2024), we adapt the embedding dimension to 1536 with 24 heads. This makes our image encoder directly comparable to DINOv2-g, counting 1.1B parameters in total. On the text side, we scale the transformer to 12 layers, with the same embedding dimension and number of heads as the image encoder.

**Data.** We leverage the WebLI dataset (Chen et al., 2023), which is a large and noisy web dataset of public images and associated alt-text containing 10B image-text pairs. We filter the dataset in successive rounds in order to enhance its quality for model training, similar to previous work in language (Gunasekar et al., 2023; Wenzek et al., 2020) and vision (Oquab et al., 2024; Parthasarathy

et al., 2023). This is critical for our model, since it is intended for off-the-shelf use in many downstream applications. First, similar to Schuhmann et al. (2022), we filter the image-text pairs based on their contents, by discarding those whose image-text similarities are low, as computed by a pretrained alignment model. Second, we filtered the resulting dataset to only keep pairs with English captions. These two initial steps result in a dataset of 1.7B images. Finally, we follow a similar curation process as previous work (Oquab et al., 2024; Parthasarathy et al., 2023) and select images that are similar enough to those in curated datasets, leveraging a pretrained model to compute image embeddings; further details on the curated datasets and filtering strategies are reported in Appendix A.3. Note that we also remove near-duplicate images from our dataset if they appeared in any of the evaluation datasets used in this paper. This process generates our main curated pretraining dataset, containing 116M image-text pairs in total.

# 4 EXPERIMENTS

## 4.1 EXPERIMENTAL SETUP

**Evaluation datasets and protocols.** Our models are evaluated on a suite of 8 tasks involving 16 datasets in total, comprising images-only or images-and-text tasks. We assess the quality of the learned representations thoroughly in a wide range of conditions, covering indoor/outdoor scenes and object-centric captures. Note that, in all evaluations, our image-text representations are kept frozen, since our goal is to assess their applicability as off-the-shelf feature extractors. We evaluate 3 dense prediction tasks, 2 holistic global image understanding tasks and 3 multimodal retrieval tasks. We introduce the tasks below, and provide further details about their evaluation protocols in the appendix.

**Semantic segmentation** is a dense task evaluated on PASCAL VOC (Everingham et al., 2010) and ADE20k (Zhou et al., 2017) datasets, using mean Intersection over Union (mIoU). We use a simple linear probe setup similar to (Oquab et al., 2024), where classes are predicted from the spatial features.

**Monocular depth estimation** aims to predict the depth value for each pixel on the image. We benchmark depth estimation on the scene-centric NYUv2 (Silberman et al., 2012), and the object-centric NAVI (Jampani et al., 2023), and we use the RMSE metric. For NYUv2, we use a linear probe setup similar to (Oquab et al., 2024), where patch tokens are concatenated to the global embedding, on top of which a linear classifier predicts among 256 quantized depth values. For NAVI, we follow (El Banani et al., 2024) and probe with the DPT (Ranftl et al., 2021) decoder.

**Surface normal estimation** is the task of densely predicting the 3D surface normal direction of each pixel, and is also assessed using NYUv2 and NAVI. We use both datasets with the setup of (El Banani et al., 2024), and report angular RMSE.

**Image classification** is evaluated on the ImageNet-1K dataset (Russakovsky et al., 2015), where we consider K-Nearest-Neighbor (KNN) and linear probe evaluations on top of the learned features. We report top-1 accuracy.

**Fine-grained and instance-level retrieval** is evaluated leveraging the Universal Embeddings Dataset (UnED) (Ypsilantis et al., 2023), which itself is a benchmark combining datasets from 8 domains: food (Food2k dataset, Min et al. (2023)), cars (CARS196 dataset, Krause et al. (2013)), online products (SOP dataset, Song et al. (2016)), clothing (InShop dataset, Liu et al. (2016)), natural world (iNat dataset, Van Horn et al. (2018)), artworks (Met dataset, Ypsilantis et al. (2021)), landmarks (GLDv2 dataset, Weyand et al. (2020)) and retail products (Rp2k dataset, Peng et al. (2020)). We report the average recall@1 (R@1) over the 8 domains, and per-domain results in the appendix.

**Image-to-text (I→T) retrieval** is assessed using the Flickr30K (Young et al., 2014), DOCCI (Onoe et al., 2024) and COCO (Chen et al., 2015) datasets, also reporting the R@1 metric.

**Text-to-image (T→I) retrieval** is similarly assessed using Flickr30K, DOCCI and COCO, with the R@1 metric.

**Zero-shot classification** is conducted on ImageNet-1K by retrieving the class text embedding closest to the each test image's embedding, following Radford et al. (2021), and using top-1 accuracy.

**Additional training dataset details.** As discussed in Sec. 3.3, we use images from a set of curated datasets as queries for mining among a large pool of web images. Following DINOv2 (Oquab et al.,

| Method | ↑ Segmentation (Pascal VOC) | ↓ Depth (NYUv2) | ↑ KNN classif. (ImageNet) | ↑ I→T retrieval (Flickr) | ↑ T→I retrieval (Flickr) |
|---|---|---|---|---|---|
| *(A) Baseline* | | | | | |
| CLIP (noisy captions) | 64.4 | 0.620 | 76.9 | 79.1 | 62.9 |
| *(B) CLIP using synthetic captions* | | | | | |
| PaliGemma captions | 74.5 | 0.544 | 70.0 | 79.8 | 60.1 |
| Both captions, sampled | 71.8 | 0.563 | 77.0 | **90.2** | 75.4 |
| Both captions, multi-text | 72.1 | 0.580 | 76.9 | 85.1 | 73.9 |
| Both captions, dual | 73.3 | 0.588 | 78.3 | 88.7 | 77.1 |
| *(C) Improved loss functions, with noisy captions* | | | | | |
| CLIP + self-dist | 70.3 | 0.589 | **79.1** | 81.5 | 67.0 |
| CLIP + self-dist + MIM | 75.9 | 0.511 | 79.0 | 82.6 | 67.6 |
| *(D) **Ours**: combining improved captions from (B) and losses from (C)* | | | | | |
| CLIP + self-dist + MIM | | | | | |
|    Both captions, dual | **79.0** | **0.478** | 78.8 | 89.2 | **77.3** |

Table 1: **Ablations for enhanced captions and improved losses**, using the ViT-B backbone on 5 representative dense, global and image-text tasks. Our final method presented in (D) achieves large gains in all tasks compared to the baseline CLIP shown in (A).

2024), we use the training sets of some of our evaluation datasets as the curated queries (details in the appendix). This leads to a web-based training dataset with 116M image-text pairs, which we use for all released models. Additionally, for the ViT-g models, we experiment with adding the training set of the Mapillary SLS dataset (Warburg et al., 2020) as-is to our training set to compensate for the lack of street-level imagery in web images, and in the absense of any alt-text we use the generated synthetic caption for training both CLS tokens. This increases the number of images in our training set to 117M in total. A similar procedure is conducted by DINOv2 for their LVD-142M dataset.

**Implementation details.** We use 1 global crop at resolution 224 and $M = 6$ local crops at resolution 98. We train the ViT-B models for 70 epochs at batch size 16k, which takes 4 days on 256 TPUv3 chips. For the ViT-g model we train for 15 epochs at batch size 16k, which takes 2 days on 512 TPUv5 chips, and results in our low-res model (**TIPS-g/14 LR**). For our high-res variant (**TIPS-g/14 HR**), we run an additional finetuning stage with global crops at resolution 448 and local crops at resolution 140, for 0.1 epochs at batch size 4k. We use only random resize crops and horizontal flips as image augmentations. Models marked with the **[rel]** prefix correspond to those which are released publicly, whose training set does not include Mapillary SLS. See the appendix for more details.

**Captioner model.** We leverage the recent PaliGemma (Beyer et al., 2024) model for image captioning. Specifically, we use the version fine-tuned on COCO, with the 224 image size version used for the core pretraining run and the 448 version for the short high-resolution fine-tuning stage.

**Compared techniques.** We strive to provide a large number of comparisons against recent work. For each existing model family, we compare against the largest instantiation up to ViT sizes "g" or "G", at about 1.8B parameters or less in the image encoder. We benchmark TIPS against a wide range of methods, from the self-supervised, weakly-supervised and supervised literature. All methods are used off-the-shelf, with frozen weights, for fair comparisons. As self-supervised methods, we compare against DINO (Caron et al., 2021), MAE (He et al., 2022), iBOT (Zhou et al., 2022) and DINOv2 (Oquab et al., 2024). As weakly-supervised methods, we compare against CLIP (Radford et al., 2021), OpenCLIP (Cherti et al., 2023), SigLIP (Zhai et al., 2023), MaskCLIP (Dong et al., 2023), SILC (Naeem et al., 2024) and EVA-CLIP (Sun et al., 2023). As a supervised method, we benchmark against the ViT-g trained on JFT-3B, as per (Zhai et al., 2022).

## 4.2 RESULTS

**Ablations.** We present in Tab. 1 ablative experiments on 5 different tasks to isolate the effect of the enhanced textual supervision and new losses, where a ViT-B backbone is used. The baseline CLIP model with the noisy web captions is presented in (A).

Part (B) of the table ablates the contribution of enhanced textual supervision. Simply replacing the web captions by PaliGemma-generated ones improves segmentation by 10.1 percentage points and reduces depth RMSE by 0.076, which are big positive gains. This shows the potential of synthetic captions for dense understanding with image-text models. However, at the same time global tasks show significant regressions, with KNN classification loss of 6.9 points. But the CLIP performance can be improved in all tasks by combining the web and synthetic captions: using our dual embedding

| Method | ↑ Segmentation | | ↓ Depth | | ↓ Normals | | ↑ Fine-grained | ↑ ImageNet classif. | |
| --- | --- | --- | --- | --- | --- | --- | --- | --- | --- |
| | PASCAL | ADE20k | NYUv2 | NAVI | NYUv2 | NAVI | retrieval (UnED) | KNN | lin |
| DINO-B | 66.4 | 31.8 | 0.555 | - | 28.4 | 28.8 | - | 77.4 | 80.1 |
| MAE-H/14 | 67.6 | 33.3 | 0.517 | - | - | - | - | 49.4 | 76.6 |
| iBOT-L/16 | 82.3 | 44.6 | 0.417 | - | 24.5 | 26.6 | - | 72.9 | 82.3 |
| JFT-3B-g/14 | 70.7 | 37.5 | 0.605 | 0.096 | 24.4 | 26.7 | 59.5 | **85.1** | **87.4** |
| DINOv2-g/14 | 83.0 | 49.0 | **0.344** | **0.054** | **20.5** | **24.0** | 62.2 | 83.5 | 86.5 |
| CLIP-L | 74.5 | 39.0 | 0.553 | 0.073 | 24.3 | 25.5 | 57.4 | 79.8 | 84.3 |
| SigLIP-SO/14 | 67.8 | 35.8 | 0.580 | 0.074 | 25.6 | 25.7 | 70.8 | 84.4 | 86.4 |
| OpenCLIP-G/14 | 71.4 | 39.3 | 0.541 | - | - | - | - | 83.2 | 86.2 |
| TIPS-g/14 LR (ours) | 82.9 | 47.8 | 0.377 | 0.061 | 23.0 | 24.5 | 71.4 | 83.6 | 86.4 |
| TIPS-g/14 HR (ours) | **83.6** | **49.9** | 0.353 | 0.058 | 21.9 | 24.2 | 68.2 | 83.3 | 86.2 |
| [rel] TIPS-g/14 LR (ours) | 82.0 | 47.4 | 0.390 | 0.063 | 23.5 | 24.7 | **71.5** | 83.6 | 86.3 |
| [rel] TIPS-g/14 HR (ours) | 83.1 | 49.4 | 0.363 | 0.059 | 22.3 | 24.3 | 68.4 | 83.2 | 86.1 |

Table 2: **Image-only evaluations for dense and global prediction tasks.** Experiments using the largest backbone available for each model variant, comparing recent self-supervised and image-text models against TIPS. Rows that are highlighted refer to image-only models that are very good at dense prediction tasks, but are by nature unable to perform text-related tasks. LR (HR) refers to our low-res (high-res) model variant. Models marked with the [rel] prefix refer to those released publicly. We also highlight the **best** and second-best number of each column. TIPS achieves the best or second-best performance in 7 out of 9 evaluations.

approach, we achieve large gains across the board. We also compare our dual approach against two other caption combination options, inspired by the ones proposed by Fan et al. (2023): "sampled", where either the web or the synthetic caption is chosen at random; or "multi-text", where both captions are matched against the same image embedding. Our dual approach performs better than other caption combinations in 3 out 5 cases and achieves competitive results on the other 2, which indicates its effectiveness.

Part (C) ablates the effect of the self-supervised losses, using web captions. The addition of self-distillation brings improvements in all tasks. This is a setup similar to SILC (Naeem et al., 2024): we confirm their findings for I→T and T→I retrieval, and additionally show that the self-distillation loss is effective for image-only tasks, notably dense ones. With our additional masked image modeling (MIM) loss, significant improvements are observed in dense tasks, while maintaining high scores in the other tasks: 5.6 points gain in segmentation and 0.078 reduction in depth RMSE.

Part (D) combines the findings of (B) and (C) to deliver very substantial improvements against the baseline CLIP setup in all tasks, notably: 14.6 points gain in segmentation, 0.142 reduction in depth RMSE, 10.1 points gain in I→T retrieval and 14.4 points gain in T→I retrieval. Additional ablations can be found in the appendix.

**Comparisons against existing general-purpose methods** are provided in Tables 2 and 3, for tasks involving images only or images and text, respectively, where results for TIPS are provided for the model before ("LR") and after ("HR") high-resolution fine-tuning. The released TIPS models are marked with the "[rel]" prefix. Overall, TIPS achieves strong results, with competitive performance across a wide range of tasks, reaching the best or second-best numbers in 13 out of the 16 reported evaluations. Compared against existing image-text methods, TIPS improves on I→T and T→I retrieval, while also achieving substantial gains in dense prediction tasks, reaching the level of DINOv2 and surpassing it in some cases. It is interesting to note that while recent image-text models have achieved excellent results in multimodal retrieval or zero-shot classification, those gains do not translate to improved features for dense understanding, whose performance lags substantially behind TIPS and self-supervised approaches. In particular, even CLIP-L, with much worse performance on image-level prediction tasks, outperforms the recent SigLIP-SO on all 6 dense evaluations. Another recent and much larger image model trained with contrastive learning, InternViT-6B (Chen et al., 2024), achieves 47.2% on ADE20k, which is much worse than our 1.1B TIPS-g model. In terms of supervised methods, the ViT-g trained on JFT-3B also performs worse on dense tasks than CLIP-L. And an even larger ViT-22B (Dehghani et al., 2023), also trained on JFT, achieves only 34.6% in ADE20k on the same setup, as reported by Chen et al. (2024). In comparison to self-supervised techniques, TIPS achieves strong results, with numbers comparable to DINOv2 in most cases and surpassing them significantly in segmentation and retrieval, while at the same time enabling multimodal tasks which cannot be performed with self-supervised methods alone. Fig. 4 shows qualitative examples for our dense feature probes. Additionally, we provide results for TIPS models distilled to smaller ViT backbones in the appendix.

**Application: Single-image to 3D.** Modern large reconstruction models rely on high-quality pre-trained image encoders to produce image tokens for an encoder/decoder transformer (Hong

| Method | ↑ I→T retrieval | | | ↑ T→I retrieval | | | ↑ ImageNet |
|---|---|---|---|---|---|---|---|
| | COCO | Flickr | DOCCI | COCO | Flickr | DOCCI | 0-shot |
| MaskCLIP-B/14 | 41.4 | 70.1 | - | 25.5 | 45.6 | - | 44.5 |
| CLIP-L/14 | 56.3 | 85.2 | 44.4 | 36.5 | 65.2 | 40.4 | 75.5 |
| OpenCLIP-G/14 | 67.3 | 92.9 | - | 51.4 | 79.5 | - | 80.1 |
| EVA-CLIP-g/14 | 68.2 | 91.6 | - | 50.3 | 78.9 | - | 79.3 |
| SigLIP-SO/14 | 70.2 | 91.0 | 27.5 | 52.0 | 75.3 | 28.4 | 83.2 |
| SILC-G/16 | 73.2 | - | - | 54.7 | - | - | **83.7** |
| TIPS-g/14 LR (ours) | 73.7 | 93.0 | 56.4 | 58.3 | 83.2 | 58.9 | 79.7 |
| TIPS-g/14 HR (ours) | **74.0** | 93.0 | **57.2** | **59.4** | **84.5** | 58.8 | 79.9 |
| [rel] TIPS-g/14 LR (ours) | 73.3 | 93.4 | 55.6 | 58.1 | 82.1 | 57.0 | 79.6 |
| [rel] TIPS-g/14 HR (ours) | **74.0** | **93.8** | 57.0 | 59.2 | 83.8 | **59.4** | 79.7 |

Table 3: **Image-text evaluations for multimodal retrieval and zero-shot classification**, where TIPS outperforms others in 6 out of 7 cases. We compare solely against weakly-supervised methods, since self-supervised ones are not naturally aligned with language. We highlight the **best** and second-best number of each column.

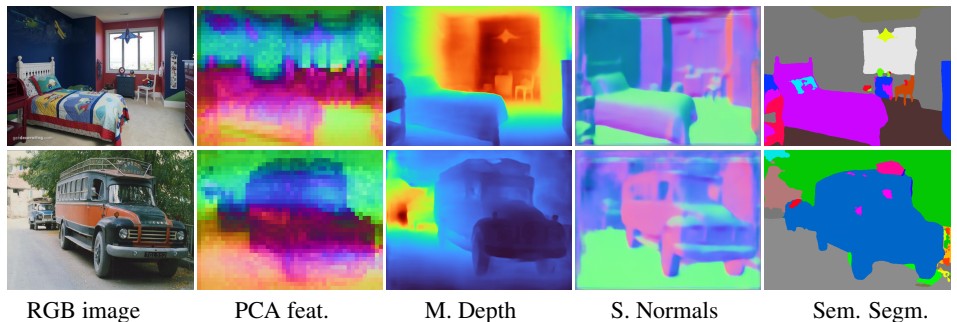

| RGB image | PCA feat. | M. Depth | S. Normals | Sem. Segm. |
|---|---|---|---|---|

Figure 4: **Qualitative dense prediction results.** For a given image (first column), we illustrate the principal components of the predicted spatial features (column 2) . Depth (column 3) and normals (column 4) are trained on NYUD, and for semantic segmentation (last column) we used the model trained on ADE20k. All dense tasks used the DPT decoder, with a frozen image encoder. More qualitative results can be found in the appendix.

et al., 2024; Wang et al., 2024). For example, LRM (Hong et al., 2024) predicts parameters of a neural rendering model from the image features of a single input image. The authors choose the ViT-based DINO encoder over more semantic-aware ones (such as CLIP) due to its knowledge of structural and texture information necessary for 3D tasks. To better understand our model's capabilities for neural 3D reconstruction, we evaluate TIPS in the LRM framework and compare DINO-B/16 to an equivalently-sized TIPS-B/14. We opt to use DINO-B/16 to follow after the original paper's implementation. Single-image to 3D results on the Objaverse (Deitke et al., 2023) dataset are presented in Tab. 4, showing that TIPS outperforms DINO as an image encoder for large reconstruction models, with enhanced novel view synthesis capabilities (0.62 increase in PSNR). Qualitative results can be found in the appendix.

| Method | ↑ PSNR |
|---|---|
| DINO-B/16 | 21.13 |
| TIPS-B/14 | **21.75** |

Table 4: **LRM novel view synthesis** using original DINO features vs. TIPS features. TIPS yields improvements with a model of similar size.

## 5 CONCLUSIONS

We introduce TIPS (Text-Image Pretraining with Spatial awareness), a new general-purpose image-text encoder. TIPS can be successfully applied off-the-shelf to a variety of computer vision tasks, enabling dense and image-level prediction, leveraging two simple and effective contributions. First, we employ existing multimodal generative models to produce high-quality synthetic image descriptions, which are used to improve contrastive learning and boost performance on dense image prediction. We propose a dual embedding approach to leverage both synthetic and noisy web captions, unlocking gains across a wide range of tasks. Second, we combine contrastive image-text learning with self-distillation and masked image modeling, incentivizing the model to learn spatially-aware representations. These two contributions are complementary and allow us to effectively scale our models to a ViT-g architecture trained on a curated dataset of 117M images. Our comprehensive experiments demonstrate strong off-the-shelf results on 8 tasks comprising 16 datasets in total, enabling a wide variety of computer vision applications which involve only images, or images and text.

ACKNOWLEDGMENTS

We would like to thank Xiaohua Zhai, Xiao Wang, Lucas Beyer, Alexey Gritsenko, Matthias Minderer, Mohamed El-Banani, Muhammad Ferjad Naeem, Austin Stone, Hagen Soltau, Jonathon Shlens, Abhijit Ogale, Ming-Hsuan Yang and Huizhong Chen for thoughtful discussions and helpful pointers to datasets, experiments and related work.

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

# A APPENDIX

## A.1 ADDITIONAL EXPERIMENTAL RESULTS

| Model | Image # Params | Text # Params | Total # Params |
|---|---|---|---|
| [rel] TIPS-S/14 HR | 21.6M | 33.6M | 55.2M |
| [rel] TIPS-B/14 HR | 85.7M | 109.6M | 195.3M |
| [rel] TIPS-L/14 HR | 303.2M | 183.9M | 487.1M |
| [rel] TIPS-SO/14 HR | 412.4M | 448.3M | 860.8M |
| [rel] TIPS-g/14 LR | 1.1B | 389.1M | 1.5B |
| [rel] TIPS-g/14 HR | 1.1B | 389.1M | 1.5B |

Table 5: **Number of parameters for all released TIPS model variants.** We release 6 different models, of 5 different sizes. For S, B, L and g model sizes, we use a fixed number of 12 layers in the text encoder; for the SO size, we use the same number of layers in both the image and text encoders.

| Method | ↑ Segmentation | | ↓ Depth | | ↓ Normals | | ↑ Fine-grained retrieval (UnED) | ↑ ImageNet classif. | |
|---|---|---|---|---|---|---|---|---|---|
| | PASCAL | ADE20k | NYUv2 | NAVI | NYUv2 | NAVI | | KNN | lin |
| [rel] TIPS-S/14 HR | 80.6 | 44.5 | 0.425 | 0.072 | 24.3 | 25.5 | 57.7 | 75.1 | 77.7 |
| [rel] TIPS-B/14 HR | 82.9 | 48.0 | 0.379 | 0.065 | 22.9 | 24.8 | 62.7 | 80.0 | 81.8 |
| [rel] TIPS-L/14 HR | 83.9 | 49.5 | 0.372 | 0.061 | 22.3 | 24.4 | 67.8 | 82.5 | 85.2 |
| [rel] TIPS-SO/14 HR | 83.7 | 49.7 | 0.362 | 0.060 | 22.0 | 24.4 | 68.6 | 83.0 | 85.7 |
| [rel] TIPS-g/14 LR | 82.0 | 47.4 | 0.390 | 0.063 | 23.5 | 24.7 | 71.5 | 83.6 | 86.3 |
| [rel] TIPS-g/14 HR | 83.1 | 49.4 | 0.363 | 0.059 | 22.3 | 24.3 | 68.4 | 83.2 | 86.1 |

Table 6: **Image-only evaluations for all released TIPS model variants**, showcasing strong performance in 5 different tasks across a variety of backbone sizes.

| Method | ↑ I→T retrieval | | | ↑ T→I retrieval | | | ↑ ImageNet 0-shot |
|---|---|---|---|---|---|---|---|
| | COCO | Flickr | DOCCI | COCO | Flickr | DOCCI | |
| [rel] TIPS-S/14 HR | 64.8 | 86.3 | 42.9 | 50.0 | 74.7 | 44.7 | 71.5 |
| [rel] TIPS-B/14 HR | 69.1 | 91.3 | 50.3 | 54.7 | 79.4 | 50.7 | 76.3 |
| [rel] TIPS-L/14 HR | 73.3 | 93.6 | 55.1 | 58.4 | 83.5 | 57.1 | 79.2 |
| [rel] TIPS-SO/14 HR | 73.4 | 94.2 | 55.9 | 58.6 | 83.8 | 58.6 | 79.1 |
| [rel] TIPS-g/14 LR | 73.3 | 93.4 | 55.6 | 58.1 | 82.1 | 57.0 | 79.6 |
| [rel] TIPS-g/14 HR | 74.0 | 93.8 | 57.0 | 59.2 | 83.8 | 59.4 | 79.7 |

Table 7: **Image-text evaluations for all released TIPS model variants**, showcasing strong performance in 3 different tasks across a variety of backbone sizes

**TIPS variants with smaller backbones.** Our objective is the development of a high-performing image-text encoder with spatial awareness. While large models trained on extensive datasets are central to this goal, practical deployment often requires smaller models. Directly training these smaller models from scratch proves suboptimal due to capacity limitations. Therefore, we adopt a knowledge distillation strategy (Hinton et al., 2015) as follows: a smaller student network is trained to reproduce the outputs of our largest model (TIPS ViT-g), particularly the outputs from the image encoder, by minimizing the difference between the teacher and student outputs for a given input set. This is achieved by adapting the TIPS training process: the image encoder teacher, initialized with the TIPS ViT-g parameters, remains frozen throughout the distillation process; an image encoder student is learned while using the original masking and local cropping strategies. The text encoder training is unchanged, being initialized from scratch. Additionally, we run a finetuning stage at higher input image resolution, as described in Sec. 4.1 for the ViT-g model variant, using the same teacher model, and initializing the student image encoder and text encoder with the weights from the previous training stage.

This training strategy is applied across a variety of backbone sizes: ViT-S, ViT-B, ViT-L (Dosovitskiy et al., 2021) and SO-400m (Alabdulmohsin et al., 2023) (we refer to the latter as simply "SO" from here on). For the text encoder scaling, we adopt the same transformer parameterization as the image encoder, but we keep the number of layers fixed at 12 (except for the SO variant, which uses its standard number of layers). As a teacher, we use our "[rel] TIPS-g/14 HR" model variant. Our preliminary studies demonstrated that this approach yields superior performance compared to training models completely from scratch, even when the student is a relatively large model like ViT-L or SO-400m.

We present the number of parameters for each of the models in Tab. 5 – all of them are released, available via our project webpage. Tables 6 and 7 present their performance across a range of tasks,

involving only images or images and text. Our distillation approach is effective at preserving high performance even for models which are much smaller than the teacher. For example, the ViT-L variant, with only 487M parameters, achieves performance across the board which is quite close to the ViT-g teacher's, with 1.5B parameters (even surpassing it in segmentation). The SO variant, at 861M parameters, surpasses the ViT-g teacher in 5 image-only evaluations and is quite close to the teacher's performance for image-text ones.

**Ablation on synthetic caption versions.** To understand in more detail the impact of synthetically-generated descriptions on different tasks, we create different variants to ablate the effect of the composition of the description. "PaliGemma object list" is created by prompting Gemini 1.5 Flash (Gemini Team Google, 2023) to take the original PaliGemma caption and produce a list of the objects that are mentioned in it, for example: "A black SUV parked in front of a building" becomes "black SUV, building". "PaliGemma main object" is created similarly to the object list version, except that it is prompted to produce only the main object in the caption, for example: "A black SUV parked in front of a building" becomes "black SUV". Results are presented in Tab. 8, using a CLIP model with a ViT-B backbone. First, note how "PaliGemma object list" already provides significant boost over the noisy captions for dense prediction tasks, which indicates that listing the multiple objects in the images, without noisy web terms, helps substantially. On top of this, the full "PaliGemma captions", including descriptions about object spatial arrangements, further improves dense understanding, notably for depth estimation. The results for "PaliGemma main object" show some improvement compared to noisy captions in depth, but not for segmentation. On I→T and T→I retrieval, "PaliGemma captions" also provide significant improvements compared against other PaliGemma caption variants.

| Captioning version | ↑ Segmentation (PASCAL) | ↓ Depth (NYUv2) | ↑ KNN classif. (ImageNet) | ↑ I→T retrieval (Flickr) | ↑ T→I retrieval (Flickr) |
|---|---|---|---|---|---|
| Noisy captions | 64.4 | 0.620 | **76.9** | 79.1 | **62.9** |
| PaliGemma captions | **74.5** | **0.544** | 70.0 | **79.8** | 60.1 |
| PaliGemma object list | 73.8 | 0.575 | 70.1 | 66.4 | 45.0 |
| PaliGemma main object | 61.9 | 0.598 | 67.7 | 8.7 | 4.7 |

Table 8: **Synthetic caption ablations** for representative evaluations, using a CLIP ViT-B model.

**Ablation on dataset versions.** We compare our final curated version of WebLI, with 116M image-text pairs in total, to 2 other versions: "raw" (10B unfiltered image-text pairs) and "EN quality-filtered" (1.7B image-text pairs filtered with pretrained alignment model and keeping only English-captioned pairs). Results are presented in Tab. 9, using a CLIP model with a ViT-B backbone, for a representative set of image evaluations. Our curated dataset helps improve performance across all of these evaluations, while being 1 or 2 orders of magnitude smaller.

| Training set | ↑ Segmentation (ADE20k) | ↓ Depth (NYUv2) | ↑ KNN classif. (ImageNet) | ↑ Fine-grained retrieval (UnED) |
|---|---|---|---|---|
| Raw (10B) | 29.1 | 0.698 | 68.4 | 45.8 |
| EN quality-filtered (1.7B) | 31.5 | 0.632 | 76.2 | 59.3 |
| Ours curated (116M) | **31.6** | **0.620** | **76.9** | **62.9** |

Table 9: **Training set ablations** for representative image evaluations, using a CLIP ViT-B model.

**Training on DataComp.** We provide additional experimental results for the CLIP baseline and for TIPS, when training on the public DataComp dataset (Gadre et al., 2023), and compare against training on WebLI (as per the main paper results). We process the DataComp dataset following the same pipeline which we used for WebLI. Starting from the DataComp-1B version (1.4B samples), which is already filtered based on image-text quality and English captions, we apply our curation process and remove near-duplicate images to those in evaluation datasets used in this paper. This leads to a DataComp dataset version with 115M image-text pairs, which is close to our WebLI-curated dataset of 116M pairs. We additionally compute PaliGemma synthetic captions for the final DataComp dataset version, which are used to train the complete TIPS method. Results are presented in Tab. 10, using a ViT-B backbone, for a representative set of image evaluations. Overall, training on these two datasets leads to very similar performance, both when training the baseline CLIP method and our TIPS technique. This is not surprising, as both datasets are collected with similar procedures, from the web. Importantly, note that significant performance gains are observed when changing from CLIP to TIPS, when training on both datasets. This indicates that the strong TIPS results can be attributed to the new training method introduced in this work, given the consistent gains.

| Method | ↑ Segmentation (PASCAL) | ↓ Depth (NYUv2) | ↑ KNN classif. (ImageNet) | ↑ I→T retrieval (Flickr) | ↑ T→I retrieval (Flickr) |
|---|---|---|---|---|---|
| *(A) CLIP* | | | | | |
| WebLI | 64.4 | 0.620 | 76.9 | 79.1 | 62.9 |
| DataComp | 64.3 | 0.620 | 76.0 | 80.4 | 64.0 |
| *(B) TIPS* | | | | | |
| WebLI | 79.0 | 0.478 | 78.8 | 89.2 | 77.3 |
| DataComp | 79.1 | 0.479 | 73.4 | 88.9 | 74.1 |

Table 10: **Comparison between training sets (WebLI and DataComp)**, using the ViT-B backbone on 5 representative dense, global and image-text tasks.

| Method | ↑ Segmentation (PASCAL) | ↓ Depth (NYUv2) | ↑ KNN classif. (ImageNet) | ↑ I→T retrieval (Flickr) | ↑ T→I retrieval (Flickr) |
|---|---|---|---|---|---|
| *(A) TIPS complete method* | | | | | |
| TIPS ViT-B (ours) | 79.0 | 0.478 | 78.8 | 89.2 | 77.3 |
| *(B) Varying masking ratio* | | | | | |
| Random, 50% | 79.3 | 0.501 | 79.1 | 90.5 | 78.0 |
| Random, 25% | 78.8 | 0.533 | 79.3 | 90.5 | 77.9 |
| *(C) Varying masking approach* | | | | | |
| Blockwise, 75% | 79.5 | 0.491 | 78.6 | 89.2 | 77.3 |
| Blockwise, 50% | 78.9 | 0.504 | 79.0 | 89.5 | 77.3 |
| Blockwise, 25% | 78.8 | 0.537 | 79.4 | 89.8 | 77.9 |
| *(D) Varying image augmentations* | | | | | |
| BYOL augs on local crop | 40.1 | 0.878 | 78.8 | 90.6 | 78.0 |
| BYOL augs local+global crop | 79.4 | 0.490 | 77.7 | 88.7 | 76.4 |
| Resize global crop | 79.4 | 0.514 | 78.2 | 89.5 | 77.5 |
| *(E) Successive CLIP + MIM* | | | | | |
| CLIP → MIM | 75.2 | 0.550 | 67.2 | 78.0 | 65.9 |

Table 11: **Ablations for design choices of self-supervised learning components**, using the ViT-B backbone on 5 representative dense, global and image-text tasks.

**Ablation on self-supervised learning components.** We present ablations on design choices for the self-supervised learning component in Tab. 11, using a ViT-B backbone model, for a representative set of image evaluations. The complete TIPS method is presented in row (A). First, we vary the masking approach (random masking or blockwise masking) and masking ratios (from 25% to 75%), as per rows (B) and (C). Note that TIPS employs random masking with 75% ratio. The results show that higher masking ratios tend to benefit dense tasks substantially, while only impacting global classification and image-text retrieval modestly. For example, depth RMSE improves significantly with higher masking ratios, with small impact on ImageNet KNN and Flickr I→T retrieval. As we aim for a spatially-aware image-text model, we find that 75% is a good trade-off. Blockwise masking performs a little worse than random masking overall, and for this reason we adopt the simpler random masking approach. Next, we vary the image augmentation approach in row (D). Our method uses only crops and flips, and we compare against the more complex augmentations introduced in BYOL (Grill et al., 2020) (color jitter, gaussian blur, solarization) applied to only the local crops as well as to both the global and local crops. We find that additional augmentations for only local crops benefits global and image-text tasks, but are significantly detrimental to local tasks, while additional augmentations for both local and global crops are detrimental for all tasks but segmentation. We also consider replacing the random square crop for the global crop with resizing to square, but find that it has mixed impact, decreasing depth performance significantly. The modest impact and performance trade-offs of complex image augmentations inform our choice to use a simple strategy of only crops and flips. Finally, in row (E) we modify the learning process to use successive CLIP then MIM training, similar to EVA (Fang et al., 2023), instead of our proposed method which leverages contrastive and self-supervised losses simultaneously. Compared to the CLIP baseline (Tab. 1 (A)), the results show improvements in dense tasks when using this strategy. This strategy also outperforms the approach that combines CLIP only with self-distillation simultaneously for dense tasks (CLIP + self-dist in Tab. 1 (C)). However, the combination of CLIP with both self-distillation and MIM simultaneously fares better across the board (CLIP + self-dist + MIM in Tab. 1 (C)).

**UnED detailed results** can be found in Tab. 12, covering all of the 8 domains in the benchmark: food (Food2k dataset, Min et al. (2023)), cars (CARS196 dataset, Krause et al. (2013)), online products

(SOP dataset, Song et al. (2016)), clothing (InShop dataset, Liu et al. (2016)), natural world (iNat dataset, Van Horn et al. (2018)), artworks (Met dataset, Ypsilantis et al. (2021)), landmarks (GLDv2 dataset, Weyand et al. (2020)) and retail products (Rp2k dataset, Peng et al. (2020)). We compare TIPS against the main competitors, outperforming SigLIP on average by 0.6 percentage point and DINOv2 with a very significant 9.2 points gain. TIPS achieves the best score in 3 domains, SigLIP in another 3 domains and DINOv2 in 2 domains.

| Method | Food2k | CARS196 | SOP | InShop | iNat | Met | GLDv2 | Rp2k | Mean |
|---|---|---|---|---|---|---|---|---|---|
| CLIP-L/14 | 46.7 | 89.8 | 63.5 | 61.1 | 72.4 | 30.8 | 36.7 | 58.3 | 57.4 |
| JFT-3B-g/14 | 56.7 | 96.9 | 64.0 | 58.0 | 75.4 | 28.2 | 27.2 | 69.6 | 59.5 |
| DINOv2-g/14 | 54.4 | 83.2 | 56.3 | 35.8 | 82.3 | 60.4 | 55.3 | 69.5 | 62.2 |
| SigLIP-SO/14 | 60.6 | 97.5 | 76.7 | 76.1 | 76.5 | 58.9 | 44.6 | 76.2 | 70.8 |
| TIPS-g/14 LR (ours) | 63.6 | 94.9 | 73.8 | 83.9 | 83.3 | 55.6 | 41.9 | 74.4 | 71.4 |
| TIPS-g/14 HR (ours) | 57.0 | 94.8 | 73.2 | 81.3 | 80.8 | 48.2 | 40.8 | 72.4 | 68.6 |

Table 12: **UnED detailed results** over the 8 fine-grained/instance-level recognition domains, measured with Recall@1. We highlight the **best** and second-best number of each column.

## A.2 ADDITIONAL IMPLEMENTATION DETAILS

Loss weight coefficients as in Sec. 3.2 are $\alpha = 1, \beta = 2$. We use the Adafactor optimizer (Shazeer & Stern, 2018) with a learning rate schedule of linear warm-up for 1.4 epochs up to 5e-4, and then linear decay down to 0 for the remaining epochs. The teacher model is updated with the EMA of the student using a momentum on a cosine schedule from 0.994 to 1. The projection heads for self-distillation and masking are identical but unshared, and consist of a 3-layer MLP, $l_2$ normalization, and a weight-normalized projection layer to a prototype dimension of 32k. We use sharpening and centering operations after the projection head to avoid collapse to either uniform or Dirac delta distributions. To center, we subtract the student scores by the EMA of their means using constant momentum 0.9. To sharpen, we set temperatures $\tau_s = \tau_s' = 0.1$, $\tau_t = 0.07$, and warm-up $\tau_t'$ along a linear schedule from 0.04 to 0.07. For TIPS-g/14 LR, we stop the training early at 15 epochs due to evaluation saturation. For TIPS-g/14 HR, we start high-resolution finetuning from the 13-epoch checkpoint of TIPS-g/14 LR.

## A.3 DATASET CURATION

Table 13 lists the high-quality datasets used to curate WebLI beyond filtering based on image-text and English language. For each target dataset, we first extract image embeddings from a pretrained model and perform k-means clustering; Table 13 includes the number of clusters chosen manually to avoid overclustering. WebLI images are assigned to its closest cluster by image embedding distance, and a probability distribution is defined over the clusters using cluster membership sizes. Then, we sample from the clusters accordingly, ignoring members that are sufficiently far from their assigned cluster center (90th percentile or above). We repeat this process for each target dataset independently and perform deduplication across all samples. We also perform deduplication with all evaluation data, which removes around 19k images.

## A.4 DETAILED EVALUATION PROTOCOLS

In this section we provide the detailed evaluation protocols of all evals used in this work. As a general remark, we use identical protocols for low-res and high-res models, without modifying the input resolution, consistent with previous work (Oquab et al., 2024). The parameters of the pretrained transformer network remain frozen throughout the evals.

### A.4.1 DENSE IMAGE TASKS

For dense prediction tasks, we evaluate the quality of the patch tokens. As is common practice, we concatenate the `[CLS]` token to each of the patch tokens. For our method, we choose the `[CLS]` token that was trained with the more spatially aware synthetic caption ($\hat{e}^g$). We attach two different types of probes to the image encoder: a simple linear layer (segmentation, depth) or a powerful DPT (Ranftl et al., 2021) decoder (depth, normals).

| Dataset Name | Dataset Size | # Clusters | # Images Sampled |
|---|---|---|---|
| PASCAL-VOC-2007-train | 2,501 | 5 | 16,935,483 |
| PASCAL-VOC-2012-train | 8,648 | 8 | 17,925,056 |
| ADE20K-train | 20,210 | 20 | 17,288,796 |
| NYU-Depth-V2-train | 24,231 | 5 | 1,004,740 |
| ImageNet-2012-train | 1,281,167 | 1000 | 17,792,551 |
| ImageNet22k-train | 12,720,275 | 1000 | 19,859,560 |
| UnED-Food2k-train | 472,349 | 100 | 4,063,232 |
| UnED-CARS196-train | 6,346 | 6 | 5,012,681 |
| UnED-SOP-train | 48,942 | 48 | 15,246,219 |
| UnED-InShop-train | 20,897 | 20 | 13,611,747 |
| UnED-iNaturalist-train | 273,929 | 188 | 10,017,768 |
| UnED-Met-train | 397,121 | 397 | 8,010,633 |
| UnED-GLDv2-train | 1,422,914 | 1000 | 12,323,999 |
| UnED-Rp2k-train | 188,724 | 50 | 561,095 |
| **Total** | **16,888,254** | - | **159,654,074** |
| **Total (after self-dedup.)** | **16,888,254** | - | **115,913,373** |
| **Total (after self+eval-dedup.)** | **16,888,254** | - | **115,894,610** |

Table 13: **Dataset curation statistics.** High-quality image datasets are used to filter WebLI. We use precomputed image embeddings to calculate image similarity between target datasets and raw WebLI data.

**Semantic segmentation.** For semantic segmentation, we train the network with images of resolution $512 \times 512$. We use batch size of 16 images, and train for 40k iterations. We attach a linear layer to the main network, and up-sample to the input resolution to apply the cross-entropy loss. At test time, we run inference on full-resolution images. We note that for DINOv2 (Oquab et al., 2024) concatenating the [CLS] token to the patch tokens consistently yields inferior results, so we report results using their original protocol that does not use the [CLS].

**Monocular depth estimation.** For NYUv2, we use the simple evaluation protocol of Li et al. (2024); Oquab et al. (2024). We train with a resolution of $480 \times 640$. Similar to segmentation evals, we concatenate the [CLS] token to the patch tokens, and we upsample by a factor of 4. We attach a linear layer and cast the prediction as classification into 256 uniformly distributed bins. Note that this version of NYUv2 is slightly different to the one used in (El Banani et al., 2024), and the numbers not directly comparable with that work.

For NAVI, we follow El Banani et al. (2024) and attach the DPT decoder (Ranftl et al., 2021) to 4 uniformly distributed layers ($l = 10, 20, 30, 40$ for ViT-g/14). We train and test on center crops of objects, with a resolution of $512 \times 512$.

For both datasets we use batch size of 8 and train for 50k iterations.

**Surface normal estimation.** We follow similar protocols to El Banani et al. (2024) for both NYUv2 and NAVI, using their data and metrics. For both datasets, we probe with the powerful DPT decoder using outputs from 4 uniformly sampled transformer blocks. For NYUv2 we train with a resolution of $480 \times 480$, and we test with full-resolution $480 \times 640$ images. For NAVI, we train and test on center crops of the objects, with a resolution of $512 \times 512$. For both datasets we use batch size of 8 and we train for 50k iterations.

### A.4.2 GLOBAL IMAGE TASKS

In global image tasks, we use the [CLS] token that corresponds to the noisy image label that often contains more fine-grained information ($\mathbf{e}^g$).

**Image classification.** We use ImageNet-1k for classification. We train a linear classifier on top of the $\mathbf{e}^g$ embedding. We use input image resolution of $224 \times 224$ for 10 epochs, using an effective batch size of 1024 (smaller batch sizes achieve identical results). We run a grid search with the search space of (Oquab et al., 2024), and we report the maximum accuracy, which is a standard practice. We also report the accuracy of soft KNN classification with 20 neighbours (Oquab et al., 2024), without training any weights.

**Fine-grained and instance-level retrieval.** We use the UnED (Ypsilantis et al., 2023) dataset for retrieval. We use an input size of $224 \times 224$, and we $l_2$-normalize the resulting $\mathbf{e}^g$ embedding. As is standard for this dataset, we run KNN with a single neighbour. As also mentioned in the main text,

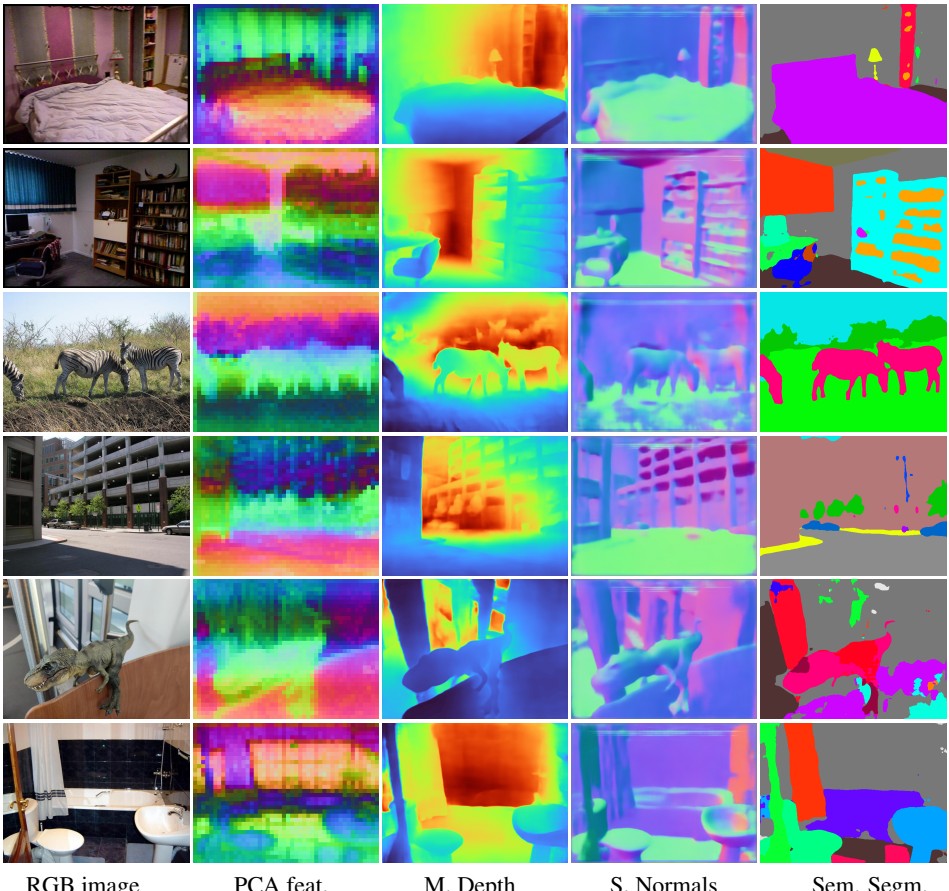

| RGB image | PCA feat. | M. Depth | S. Normals | Sem. Segm. |

Figure 5: **More qualitative results for dense prediction tasks.** For a given image (first column), we illustrate the principal components of the predicted spatial features (column 2). Depth (column 3) and normals (column 4) are trained on NYUD, and for semantic segmentation (last column) we used the model trained on ADE20k. All dense tasks used the DPT decoder, while keeping the image encoder weights frozen.

UnED consists of 8 datasets of different domains. We report recall@1 as the final aggregated result, and report results per domain in Sec. A.1.

### A.4.3 MULTIMODAL RETRIEVAL TASKS

**Image-to-text (I→T) and text-to-image (T→I) retrieval.** We use the [CLS] token that was trained with the synthetic caption ($\hat{e}^g$) as the image representation. We prefer synthetic captions because they tend to describe image content more comprehensively than noisy web captions, making them better aligned with the image-text retrieval evaluation datasets used in this work: Flickr30k (Young et al., 2014), DOCCI (Onoe et al., 2024) and COCO (Chen et al., 2015). Our text encoder handles a maximum of 64 tokens. Texts longer than this limit are truncated, which happens more frequently in DOCCI. Notably, the text encoder of SigLIP has a small maximum token length of 16, which likely contributes to its relatively low performance in DOCCI.

**Zero-shot classification.** We use the [CLS] token that corresponds to the noisy web caption, which often contains more fine-grained information ($e^g$), as image embedding. We adhere to the established protocol initiated by Radford et al. (2021), which utilizes 80 context prompts to transform each ImageNet1k class into 80 distinct texts. The class embedding is then formed by taking the average of the embeddings from these 80 texts. Classification of the image is achieved by retrieving the nearest class embedding.

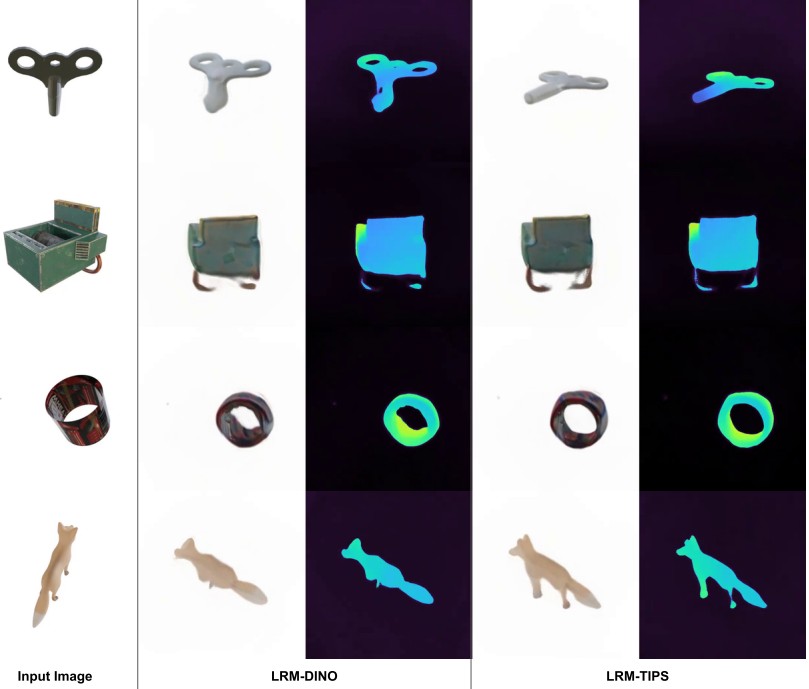

Figure 6: **Qualitative results on novel view synthesis** from LRM (Hong et al., 2024) trained on top of DINO-B/16 and TIPS-B/14. The input image is used to generate parameters for a neural rendering model, after image encoding. We visualize RGB and depth images rendered with *the same camera parameters* (intrinsics and extrinsics) using NeRF-style ray marching. We can observe that LRM-TIPS is able to predict the geometry of the captured object with a higher degree of accuracy, as indicated by the gain in PSNR from Tab. 4.

### A.4.4   3D VISION TASKS.

**Single-image to 3D.** We use the baseline LRM (Hong et al., 2024) method and follow the architectural and training details of the original paper closely unless otherwise mentioned. We obtain images from Objaverse (Deitke et al., 2023) by sampling random rotations around each object and rendering with uniformly sampled focal lengths $\in [512, 1024]$ and a nominal distance of $0.4$. Cameras are oriented towards the origin and objects are centered at the origin. For each entry in Objaverse, we render reference views for training and novel target views for evaluating view synthesis. We freeze the image encoder to extract patch features and directly compare the paper baseline (DINO-B) with ours at the ViT-B/14 scale, for a fair comparison. Figure 6 shows qualitative results of LRM trained on top of DINO-B/16 and our TIPS-B/14. We show the input single image used to generate neural rendering model parameters, along with RGB and depth images rendered via NeRF-style ray marching (Mildenhall et al., 2020).

### A.5   ADDITIONAL QUALITATIVE RESULTS

Figure 5 illustrates further qualitative results of the features and outputs obtained by our method for dense features, probing the image encoder features. In Figure 6 we show the qualitative improvement when substituting the default DINO-B with the TIPS-B model for novel view synthesis.

### A.6   DUAL EMBEDDING ATTENTION MAPS

Our model generates two distinct image embeddings, an object-centric embedding ($\mathbf{e}^g$) and a spatially-aware embedding ($\hat{\mathbf{e}}^g$). These embeddings are designed to focus on different aspects of the image: the object-centric embedding captures fine-grained information about the main object, while the spatially-aware embedding encodes compositional information about multiple objects in the scene. To validate these roles, we examine whether their attention maps are distributed consistently with

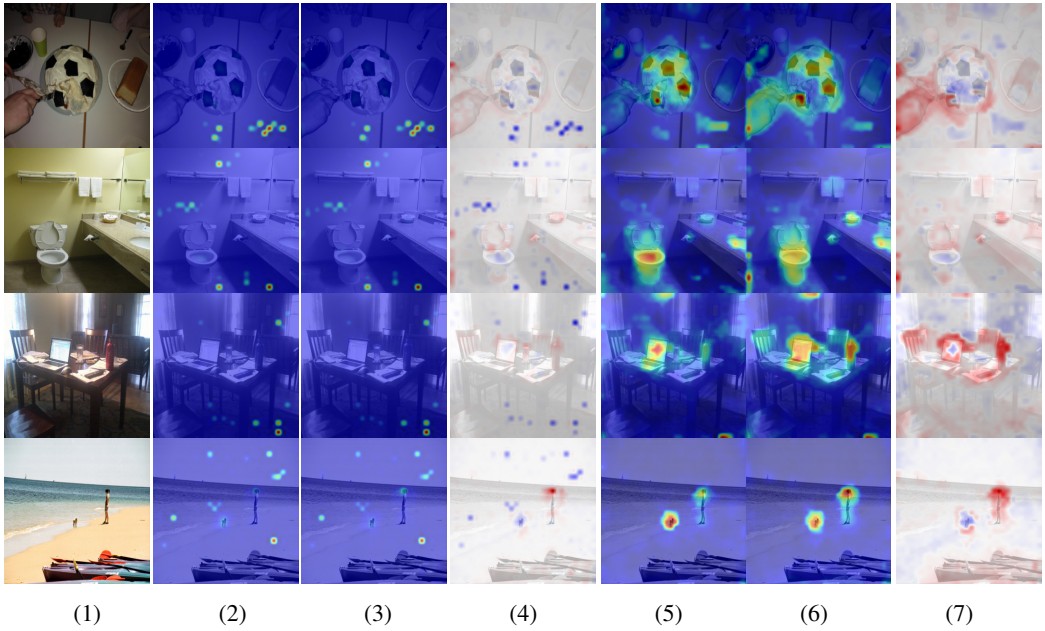

|     |     |     |     |     |     |     |
| (1) | (2) | (3) | (4) | (5) | (6) | (7) |

Figure 7: **Visualization of the attention maps** for the object-centric image embedding ($\mathbf{e}^g$) and the spatially-aware image embedding ($\hat{\mathbf{e}}^g$). **Column 1**: Input images sampled from the MSCOCO test set. **Column 2**: Attention maps of the object-centric image embedding. **Column 3**: Attention maps of the spatially-aware image embedding. While the attention maps in Columns 2 and 3 are visually similar, they exhibit meaningful differences, as highlighted in Column 4. The attention peaks in these maps correspond to global proxy patches, as reported in Darcet et al. (2024), which encode global image information. **Column 4**: Differences between the two attention maps. Red regions indicate areas where the spatially-aware embedding assigns more attention, while blue regions indicate areas where the object-centric embedding assigns more attention. The spatially-aware embedding focuses less on global proxy patches and distributes attention more broadly across different objects. **Columns 5–6**: Filtered attention maps for the object-centric embedding and the spatially-aware embedding, respectively. Filtering was performed using a median filter with a kernel size of 3 on the 32×32 resolution attention maps, which dilutes the effect of the global proxy patches and highlights image regions which are attended to more prominently. **Column 7**: Difference between the two filtered attention maps. Red regions represent areas with greater attention from the spatially-aware embedding, while blue regions represent areas with greater attention from the object-centric embedding. The spatially-aware embedding distributes attention more evenly across multiple objects in the scene, whereas the object-centric embedding focuses more on the main object.

these expectations. We highlight two key findings: (1) as observed in Darcet et al. (2024), a few sparse patches in low-informative background areas receive high attention, functioning as "global proxy patches" that encode global information; (2) the spatially-aware embedding attends less to these global proxy patches and distributes its attention more evenly across multiple objects in the image, which agrees with our expectations.

We randomly selected 500 images from the MSCOCO test set to analyze the attention weights of $\mathbf{e}^g$ and $\hat{\mathbf{e}}^g$ over the patch tokens. Example attention maps for both embeddings are shown in Figure 7, columns 1–3. Since $\mathbf{e}^g$ and $\hat{\mathbf{e}}^g$ are not far apart in the embedding space (cosine distance around 0.15 generally), their attention maps are visually similar at a coarse level. However, their differences are highly informative.

First, both embeddings share a common pattern: sparse patches in low-informative background areas—global proxy patches—receive the most attention, contributing significantly to the global image embedding. Second, the spatially-aware embedding ($\hat{\mathbf{e}}^g$) differs by assigning less attention to these global proxy patches and distributing its focus more broadly across patches corresponding to different objects in the image. These differences are further visualized in Figure 7, column 4, where red indicates where the spatially-aware embedding focuses more and blue where the object-centric one focuses more. We further analyzed the attention maps by filtering out the peak attention on global

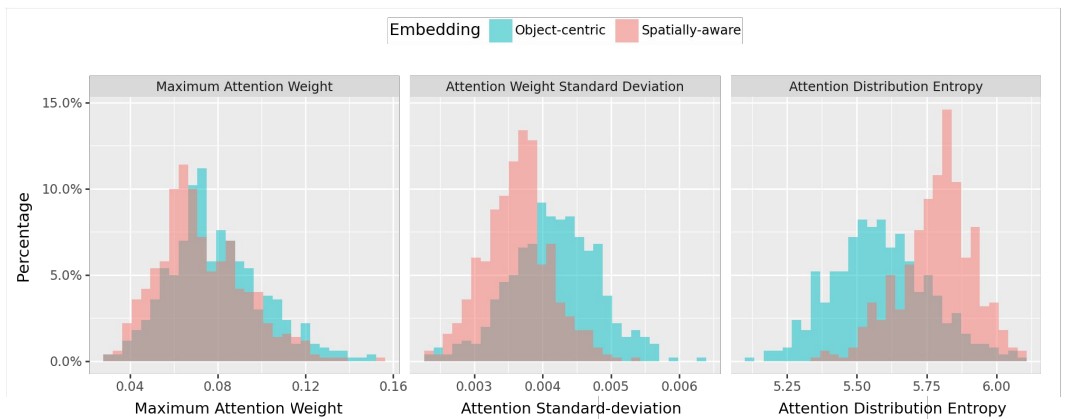

Figure 8: **Distributions of statistical metrics for the attention maps** of the object-centric embedding ($e^g$) and the spatially-aware embedding ($\hat{e}^g$), computed over a sample of 500 images randomly selected from the MSCOCO test set. The green bars represent the attention of the object-centric embedding, while the red bars represent the attention of the spatially-aware embedding. The results indicate that the spatially-aware embedding has lower maximum attention weights, lower standard deviations, and higher entropy values, demonstrating that its attention is less concentrated on global proxy patches and more evenly distributed across the image compared to the object-centric embedding.

proxy patches using a simple median filter with a kernel size of 3, applied to the 32×32 resolution maps (Figure 7, columns 5–7). This filtering highlights that the spatially-aware embedding allocates attention more broadly to multiple objects, while the object-centric embedding remains focused primarily on the main object.

To quantify the differences between the two embeddings' attention maps, we measured three metrics: the maximum attention weight, the standard deviation of the attention weights, and the entropy of the attention distribution. Specifically, for each of the 500 test images, we computed these metrics for both embeddings' attention maps. The maximum attention weight, typically found on a global proxy patch, indicates how concentrated the attention is on these patches. A higher value reflects a greater focus on global proxy patches. Similarly, the standard deviation of the attention weights reflects the spread of attention; a more evenly distributed attention map has a lower standard deviation. Finally, treating the attention weights as a probability distribution, we computed the entropy, where higher values indicate a more evenly distributed attention. The distributions of these metrics for the two embeddings are shown in Figure 8. The results reveal that the spatially-aware embedding ($\hat{e}^g$) exhibits lower maximum attention weights, lower standard deviations, and higher entropy values, confirming that its attention is more evenly distributed and less focused on global proxy patches compared to the object-centric embedding ($e^g$).

These findings align with our observations that the object-centric embedding performs better on global tasks, while the spatially-aware embedding excels at tasks requiring spatial awareness. Overall, the complementary attention patterns of the two embeddings validate the effectiveness of our dual embedding design in capturing diverse types of image information.

