# OpenReview forum: "TIPS: Text-Image Pretraining with Spatial awareness"
_ICLR.cc/2025/Conference — ICLR 2025 Poster_

### Official Review · Reviewer_wggG · 2024-11-01

**Soundness:** 2
**Presentation:** 3
**Contribution:** 2
**Rating:** 6
**Confidence:** 4

**Summary:**

This paper addresses dense and global vision tasks by enhancing textual supervision and integrating contrastive image-text learning with self-supervised techniques. The method combines noisy web captions with synthetically generated captions to improve spatial awareness and applies masked image modeling to promote coherence in spatial understanding. As a result, the model demonstrates robust performance across various tasks without the need for fine-tuning, showcasing its general-purpose applicability in both image-only and image-text applications.

**Strengths:**

1.	The paper is well-structured and clearly articulated, with detailed experimental records. By cleaning and constructing a high-quality dataset and incorporating self-supervision, methods such as dual captioning and masked image modeling enable the model to achieve significant (albeit incremental) advancements in dense prediction tasks.
2.	The trained model demonstrates strong generalizability across multiple tasks, indicating its broad applicability in vision tasks.
3.	The paper includes a substantial amount of experimental comparisons and work.

**Weaknesses:**

1.	The work presents only a limited amount of novelty. The main critique lies in the lack of significant innovation. The paper largely repurposes existing techniques like synthetic captioning and contrastive learning, and while the results are solid, they do not represent a substantial leap forward in the field.
2.	The improvements over existing models such as CLIP and DINOv2 are incremental, and the performance gains are sometimes marginal or context-specific. The originality in combining these techniques does not feel transformative.
3.	More detailed ablation studies focusing on the contribution of each component (e.g., the specific impact of spatial coherence from the captions) could strengthen the claim of novelty.

**Questions:**

1.	Authors are suggested to add detailed ablation results to isolate the impact of the synthetic captions on different spatial tasks.
2.	Have you considered alternative ways of introducing spatial awareness besides synthetic captions and masking?

---

> ### Author Response · Authors · 2024-11-20
> **Rebuttal comment to wggG**
>
> We thank the reviewer for the detailed comments. We are encouraged by the many positive remarks and will try our very best to address the reviewer’s concerns.
>
> In the following, we provide detailed responses to the weaknesses and questions from the reviewer:
>
> **W1**) *Potentially limited novelty.*
> First of all, we are glad to see the reviewer acknowledge solid results, despite the concern. While it is true that we are inspired by previous work’s explorations on synthetic captions and contrastive/self-supervised learning, we would like to highlight a few points:
> - It was previously not known that synthetic captions could benefit spatial understanding tasks, and the improvements are very significant (Tab 1). To the best of our knowledge, ours is the first paper to demonstrate this.
> - We propose a novel dual image-text embedding learning technique, which shows strong results by combining synthetic and alt-text captions in the right way.
> - We are the first to combine contrastive image-text learning with self-distillation and masked image modeling at the same time, showing that they can provide complementary strengths with positive synergy, and lead to outstanding experimental results in a broad range of tasks.
>
> **W2**) *Incremental gains over CLIP and DINOv2.*
> First, we would like to highlight that our main goal is to design a general-purpose method achieving strong performance across both spatial understanding and image-text tasks. CLIP and DINOv2 are disjoint models that lack capabilities on spatial understanding and image-text, respectively, while ours is the first model shown to provide strong results in both of these tasks.
> Second, we highlight some strong performance improvements of our method compared to recent work:
> - TIPS outperforms DINOv2 in fine-grained retrieval (UnED) by 9.2% absolute (Tab 2). Note that TIPS has close or better performance on other evals compared to DINOv2.
> - TIPS outperforms the same-size ViT-g EVA-CLIP in absolute terms by 5.8% on COCO I→T, 9.1% on COCO T→I, 1.4% on Flickr I→T, 5.6% on Flickr T→I, as per Tab 4 (*relabeled to Tab 3 in latest version*). These are substantial improvements upon recent work.
>
> **W3**) *Ablation study on impact of spatial coherence from the captions.*
> Thank you for the suggestion. We would like to point out that this ablation study is already provided in Tab 1, showing that spatial tasks (segmentation and depth) improve significantly when using synthetic captions. Simply replacing the web captions by PaliGemma-generated ones improves segmentation by 10.1 percentage points and reduces depth RMSE by 0.076, which are big positive gains (compare Tab 1 (A) vs Tab 1 (B) “PaliGemma captions”).
> Additionally, we provide more synthetic caption ablations in Table 5 (appendix), which help understand which components in the synthetic captions help spatial tasks – e.g., segmentation benefits significantly from listing the different objects in the image, while depth obtains a substantial boost when spatial relationships of scene content are described in the caption.
> We hope that these results help alleviate the reviewer’s concern but are happy to continue the discussion in case there is any additional feedback.
>
> **Q1**) *Ablations isolating the impact of synthetic captions on spatial tasks.*
> This is the same as **W3**, see answer above.
>
> **Q2**) *Alternative ways of introducing spatial awareness besides synthetic captions and masking.*
> Yes, additional ways could include the use of dense annotations such as boxes and masks. Since these are generally expensive to collect, one possibility would be to use high-confidence boxes and masks produced by state-of-the-art off-the-shelf models as pseudo ground-truth annotations. We can also leverage synthetic captions of boxed regions to obtain richer grounded supervision, e.g. “a yellow city bike” instead of simply “bike” (which is usually what one would obtain with standard class names). In our paper, though, we aimed to keep our method as simple as possible without requiring any additional expensive annotation, which could complicate the model design, but this could be a fruitful research direction.
> Additionally, we plan to explore modifications to our training strategy to enhance spatially-grounded multimodal learning. For example, one can leverage the text embedding to find the most text-aware patches in the image, and preferably mask them to incentivize the model to learn better visual representations.

---

> > ### Author Response · Authors · 2024-11-22
> >
> > We have now uploaded a new PDF version with the latest modifications to the manuscript, as per suggestions from all reviewers.
> >
> > We believe that all concerns from the reviewer have been addressed in the previous comment in this thread, and we thank the reviewer for the attention here. We sincerely hope that these notes can help the reviewer finalize the assessment of our work. We continue to be available for discussions in case any further clarifications can be helpful.

---

> > > ### Author Response · Authors · 2024-11-27
> > >
> > > We have now uploaded the final revised PDF version with the latest modifications to the manuscript.
> > >
> > > We would like to point out that we have included additional ablation experiments, in addition to the ones isolating the impact of synthetic captions on spatial tasks (which we discussed in the above answer to **W3**). While the original ablations (Tab 1, Tab 5, Tab 6) assess the contribution of each component in the TIPS method, the new ablations provided in Tab 7 and Tab 8 consider TIPS training on a different dataset and provide a detailed study on self-supervised learning components. We hope that these additional results help the reviewer’s assessment of our work.
> > >
> > > Thanks once again for your attention. We continue to be available for discussions in case any further clarifications can be helpful.

---

> > ### Comment · Reviewer_wggG · 2024-12-03
> >
> > Thanks for the resonse. Most of my concerns have been addressed. I have improved my score, yet I still hope to see more extensive experiments.

---

> > > ### Author Response · Authors · 2024-12-03
> > >
> > > Thanks for the reply. We are glad to see that reviewer’s concerns were addressed and that the score was improved, leading to an acceptance recommendation.

---

### Official Review · Reviewer_BGwY · 2024-11-01

**Soundness:** 3
**Presentation:** 4
**Contribution:** 3
**Rating:** 6
**Confidence:** 4

**Summary:**

This paper targets integrating the paradigms of both image-text representation learning and self-supervised learning to improve the spatial awareness of the former. For the SSL branch, the authors leverage the DINO V2 (iBOT) pre-training method; for the image-text branch, they propose the dual image-text embedding technique that learns from both noisy and sythetic captions while harnessing the distribution gap between two types of captions. The effeciveness of the proposed method is evaluated on several image-level multimodal tasks and comprehensive dense image prediction tasks.

**Strengths:**

- This paper is well written.

- The experiments on dense image prediction tasks are comprehensive and promising, outperforming DINO V2 on several tasks.

- Improving the spatial awareness of image-text representation learning is an important direction, combining DINO v2 and CLIP, where both are foundational works in their respective fields, is intuitive and promising.

**Weaknesses:**

- The technical contributions are limited. The proposed method is a combination of existing methods, with the dual embedding technique being the only novel contribution. Nonetheless, I'm okay with this, since the proposed model effectively and adequately solves model's spatial awareness limitation.

- As claim in Line 300:
>Our method is the first to demonstrate that combining contrastive image-text learning with self-distillation and masked image modeling leads to improvements across many tasks

However, both integrating CLIP with self-distillation and masked image modeling[1][2] have been proposed before. And this paper lacks a further discussion against these works.

- Since this is a multimodal model with spatial awareness, only I$\rightarrow$T and T$\rightarrow$I retrieval tasks are not enough to evaluate the model's fine-grained spatial awareness under multimodal settings. Including more experiments like open-vocabulary segmentation would be beneficial.

reference:

[1] Maskclip: Masked self-distillation advances contrastive language-image pretraining.  CVPR 23.

[2] Scaling Language-Image Pre-Training via Masking. CVPR 23

**Questions:**

- As the motivation of this paper is to bridge the gap between image-text representation learning and SSL, although the ablation studies are provided, this paper lacks an in-depth analysis on how the two paradigms interact with each other. For example, how the SSL design choices such as augmentations (mask ratio, etc.) affect the image-text representation learning.

- The idea of dual embedding is interesting. I'm curious about the different roles of the two embeddings, and how they interact with the network. Could the authors provide more empirical analysis on this? For example, visualization of the attention maps of the two different $[CLS]$ to see their focus areas.

---

> ### Author Response · Authors · 2024-11-20
> **Rebuttal comment to BGwY**
>
> We thank the reviewer for the detailed comments. We are encouraged by the positive evaluation of our work.
>
> Some of the weaknesses and questions raised in the review suggest the need for additional experimental studies. We are currently working hard on experiments which could provide results to help alleviate the concerns, and will send an update in this comment thread once the results are available.
>
> In the following, we provide detailed responses to the weaknesses and questions from the reviewer:
>
> **W1**) *Potentially limited technical contributions.*
> Firstly, we are glad to see the reviewer acknowledge that the proposed model effectively and adequately solves spatial awareness limitations. Our work does leverage learnings from previous work in representation learning, however we would like to highlight a few points:
> - It was previously not known that synthetic captions could benefit spatial understanding tasks, and the improvements are very significant (Tab 1). To the best of our knowledge, ours is the first paper to demonstrate this.
> - As the reviewer points out, we propose a novel dual image-text embedding learning technique. To emphasize, our experiments show that combining synthetic and alt-text captions in the right way helps with a variety of downstream applications, for both dense or image-level prediction.
> - Our method is the first to combine image-text contrastive learning with masked image modeling and self-distillation at the same time, showing that they can provide complementary strengths with positive synergy, resulting in outstanding experimental results in a broad range of tasks.
>
> **W2**) *Potential issues with claim in Line 300: missing FLIP/MaskCLIP citations?*
> We thank the reviewer for the detailed comments on this point, which will help us improve our paper and better position our work against previous methods. Let us discuss in detail:
> - The reference “Scaling Language-Image Pre-Training via Masking” pointed out by the reviewer corresponds to the FLIP paper, which is already discussed in the submission (see reference Li et al., 2023). FLIP has a very different goal from ours, since they proposed to combine contrastive learning with masking without any reconstruction loss, aiming only at efficient language-image training (no spatial awareness goal). We believe that the current version of our manuscript provides sufficient discussion regarding FLIP, but we are open to any additional suggestions from the reviewer on this point.
> - Regarding the “MaskCLIP” reference: indeed, we have missed it in the submitted version of the paper, and we will fix this. We will upload a new PDF version of the paper in the next few days with this reference included, including relevant discussion. In terms of differences from our TIPS method compared to MaskCLIP, we would like to emphasize: i) our approach goes beyond masked image modeling to also include self-distillation losses, which we show important via ablations; ii) we show the power of synthetic captions for spatial understanding, which is not an aspect studied in their work; iii) we aim for off-the-shelf usage for many vision tasks, which is different from their goal (their dense prediction results are mainly in the setup of full model fine-tuning, covering only a small number of dense tasks). Additionally, we can directly compare some experimental results of our method against MaskCLIP: while our ViT-B model achieves 89.2% on Flickr I→T retrieval (Tab 1), MaskCLIP’s ViT-B achieves only 70.1% (Tab 5 in their paper). And on Flickr T→I, we achieve 77.3%, compared to MaskCLIP’s 45.6%. This shows that our full training recipe for TIPS tends to achieve significantly better results than MaskCLIP.
>
> **W3**) *Experiments on fine-grained spatial awareness under multimodal settings.*
> We are working on this experiment currently and will report results in this comment thread once they are available.
>
> **Q1**) *How SSL design choices affect image-text representation learning.*
> We are working on this ablation experiment currently and will report results in this comment thread once they are available.
>
> **Q2**) *Visualization of the attention maps of the two different [CLS] to see their focus areas.*
> We are working on this experiment currently and will report results in this comment thread once they are available.

---

> ### Author Response · Authors · 2024-11-22
>
> We have now uploaded a new PDF version with the latest modifications to the manuscript, as per reviewer suggestions. We would like to point out the following changes:
> - Ablation on SSL components (**Q1** in the previous comment): Table 7 (*relabeled to Tab 8 in the latest version*) was added to the appendix, with the associated paragraph “Ablation on self-supervised learning components”. These experiments report results varying the masking approach and ratio for the masked modeling component of TIPS. Additional ablations will be included in the next few days.
> - MaskCLIP discussion (**W2** in the previous comment): included the MaskCLIP reference, which was previously missing. Discussed it in Related Work (section 2) and section 3.2, in relation to the proposed method. Added MaskCLIP results in Tab 3.
>
> We are continuing to work on the additional experiments suggested by the reviewer and will report back once they are ready.
>
> Thanks once again for your attention. We continue to be available for discussions in case any further clarifications can be helpful.

---

> > ### Author Response · Authors · 2024-11-26
> >
> > We have now uploaded a second revised PDF version with the latest modifications to the manuscript.
> >
> > We have included additional experiments as per the reviewer’s request:
> > - Dual embedding attention visualization (addresses **Q2**): see new appendix section A.6, with detailed visualizations and discussions on the roles of the two embeddings, which corroborate our intuitions that the two embedding heads focus on different aspects of the image.
> > - Additional ablation on SSL components (addresses **Q1**): the new Table 8 now reports ablations considering additional image augmentations, and related discussion was added to the paragraph “Ablation on self-supervised learning components”.
> >
> > We are continuing to work on the remaining experiment suggested by the reviewer (fine-grained spatial awareness under multimodal settings), and will report back once it is ready.
> >
> > Thanks once again for your attention. We continue to be available for discussions in case any further clarifications can be helpful.

---

> > > ### Author Response · Authors · 2024-11-27
> > >
> > > We would like to provide an update on the remaining experiment suggested by the reviewer (fine-grained spatial awareness under multimodal settings, **W3** above). We thank the reviewer again for suggesting this additional experiment.
> > >
> > > We evaluated TIPS for zero-shot semantic segmentation, i.e. the similarity of the image patch tokens with the query class text token (grounding) for the task of semantic segmentation. For a fair comparison, and since we are using a different framework, we re-implemented the evaluation protocol of TLC* [B]. We evaluate the raw features, without any training, or post-processing**.
> > >
> > > We use a TIPS model with a global average pool (GAP) head for the image embedding***. We evaluate on two different datasets: PASCAL VOC (VOC20) and ADE20k (A150). We compare TIPS to the state-of-the-art [C], and [A] which is the pioneering work in this area.
> > > TIPS achieves an IoU of 78.6% on VOC20, better than both [C] (77.5%) and [A] (53.7%). On A150, TIPS achieves mIoU of 17.8%, better than [A] (10.8%) and slightly below [C] (19.3%).
> > >
> > > \* The evaluation protocol of [B] uses an average of 80 different prompts for embedding each class (such as `this is an image of {}`). It uses an input size of 448x448, with a sliding window of 224x224).
> > >
> > > ** Post-processing methods significantly improve the performance of zero-shot semantic segmentation. For example, [A] trains on pseudo-ground-truth spatial labels, and filters out non-existing classes with a process called prompt denoising, while [B] uses post-processing with Pixel-Adaptive Mask Refinement (PAMR). While TIPS can orthogonally benefit from these techniques, in our experiments we evaluate the raw representations produced by TIPS.
> > >
> > > *** While we use the CLS tokens for embedding the real and synthetic captions, these are not a direct function of the output patch embeddings. Therefore, the vanilla patch embeddings are not necessarily grounded to the text. We tried different approaches for improving this, including using the values of the last encoder block [A, C], or using different embedding heads, such as MAP or GAP. Using a simple global average pooling (GAP) worked the best out of all variants.
> > >
> > > References:
> > > - [A] Zhou et al., Extract Free Dense Labels from CLIP, ECCV 2022.
> > > - [B] Cha et al., Learning to Generate Text-grounded Mask for Open-world Semantic Segmentation from Only Image-Text Pairs, CVPR 2023.
> > > - [C] SILC: reference “Naeem et al., 2024” in our paper.

---

> > > > ### Comment · Reviewer_BGwY · 2024-11-29
> > > >
> > > > Thanks for the detailed rebuttal. Most of my concerns have been effectively addressed. The paper does a good job of enhancing CLIP with SOTA SSL techniques and synthetic captions. While I still believe that the technical contributions of combining DINO V2 with CLIP may be somewhat limited, I have decided to maintain my original score and am inclined to recommend the paper for acceptance.

---

> > > > > ### Author Response · Authors · 2024-11-29
> > > > >
> > > > > Thanks for the reply. We are glad to see that reviewer’s concerns were effectively addressed, and that the reviewer recognizes the "good job" of the paper, leading to an acceptance recommendation.

---

### Official Review · Reviewer_FXGp · 2024-11-03

**Soundness:** 4
**Presentation:** 4
**Contribution:** 4
**Rating:** 8
**Confidence:** 3

**Summary:**

This paper introduces a novel pretrained image-text encoder with spatial awareness which is effective in a variety of downstream computer vision tasks. To achieve this, the author first employs pretrained multimodal generative models to generate high-quality synthetic image descriptions and develops a dual embedding approach that leverages both synthetic and noisy web captions in training. Additionally, contrastive image-text learning, coupled with self-distillation and masked image modeling, is introduced to encourage the model to learn spatially aware representations. Experiments conducted on eight downstream tasks validate the effectiveness of the proposed method.

**Strengths:**

1. The author proposes an effective approach that enhances the utility of both synthetic and noisy web captions in training. They also introduce contrastive image-text learning with self-supervised masked image modeling, which effectively encourage the learning of spatial coherence.
2. The author conduct a variety of experiments in 8 downstream tasks demonstrate the effectiveness of its spatial-aware text-image encoder.

**Weaknesses:**

The formatting of the paper needs improvement and there are  a lot of empty spaces around fig1 and fig2.

**Questions:**

Will the pretrained model and the curation dataset with synthetic captions be released?

---

> ### Author Response · Authors · 2024-11-20
> **Rebuttal comment to FXGp**
>
> We thank the reviewer for the detailed comments. We are encouraged by the positive evaluation of our work.
>
> Here we respond to the weakness and the question raised by the reviewer:
>
> **W1**) *Formatting.*
> We agree that there is room for improvement on the paper formatting, in particular around Figures 1 and 2, as pointed out by the reviewer. We will make changes accordingly and upload a new PDF version of the paper in the next few days.
>
> **Q1**) *Model/data release.*
> We are planning to release the pretrained model together with the final version of the paper. We are currently following the model release process required by our organization and expect that all approvals will be obtained in time.
> Regarding the dataset: our curated dataset is part of a much larger corpus of images that has not yet been publicly released. Therefore, unfortunately, our organization prohibits its release.

---

> > ### Author Response · Authors · 2024-11-22
> >
> > We have now uploaded a new PDF version with the latest modifications to the text, as per reviewer suggestions. We would like to point out that the formatting changes suggested by the reviewer were adopted (**W1** in the previous comment): we fixed the spacing issues around figures 1 and 2, as requested. We additionally improved formatting as follows: (i) rearranging the placement of some tables and fixing the numbering of Tables 3 and 4; (ii) improving spacing in the new Table 3 (compare to previous Table 4); (iii) enhancing formatting of Tab 2 to guide the reader more effectively over all results.
> >
> > We believe that all concerns from the reviewer have now been addressed, and we thank the reviewer for the attention here. We continue to be available for discussions in case any further clarifications can be helpful.

---

### Official Review · Reviewer_6Wzj · 2024-11-04

**Soundness:** 3
**Presentation:** 3
**Contribution:** 3
**Rating:** 6
**Confidence:** 4

**Summary:**

This paper presents a spatial-aware text-image pre-training method that combines contrastive image-text learning with self-supervised masked image modeling. Besides, the method proposes to combine the noisy web captions and synthetic captions that are more helpful to learn spatially aware representations. The method is evaluated on both zero-shot classification and dense prediction tasks.

**Strengths:**

- The paper presents a solid work on text-image pre-training: a large-scale synthetic caption dataset is created, the method is evaluated on both classification and dense prediction tasks, and extensive experimental studies are conducted for ablation studies and analyses.

- The results look good. The proposed method can achieve good dense prediction and classification/retrieval performance simultaneously. Ablation results provided in the paper may be helpful for developing new text-image models.

**Weaknesses:**

- The general idea of combining contrastive image-text learning and masked image modeling is not new. Previous work like EVA-CLIP [r1] has already show that MIM can improve the spatial awareness or locality of CLIP features and improve CLIP performance. The core different between TIPS and the line of work is to combine MIM and CLIP successively or simultaneously. I think simultaneously perform the two tasks may be better to preserve the spatial awareness/locality, but it may also make the training more costly, or possibly unstable. It would be better to provide a comparison/analysis on the pros and cons of the two strategies.

[r1] EVA-CLIP: Improved Training Techniques for CLIP at Scale

- The study use the proprietary WebLI dataset to train the model. Is it possible that the improvements over previous methods mainly come from better data sources?  How about the results if both the proposed model and the baseline use publicly available datasets like LAION, COYO or DataComp.

**Questions:**

Please refer to my comments above.

---

> ### Author Response · Authors · 2024-11-20
> **Rebuttal comment to 6Wzj**
>
> We thank the reviewer for the detailed comments. We are encouraged by the positive evaluation of our work.
>
> The two weaknesses raised in the review suggest the need for additional experimental studies. We are currently working hard on experiments which could provide results to help alleviate the two concerns, and will send an update in this comment thread once the results are available.
>
> In detail, the two weaknesses were:
>
> **W1**) *Comparing against the EVA method of combining contrastive image-text learning and masked image modeling.*
> The reviewer mentions EVA-CLIP, and by that we understand the suggestion to compare against the sequential method of i) contrastive, then ii) MIM reconstruction, which was originally presented in the first EVA paper (reference “Fang et al., 2023” in the paper). We are working on experiments to compare this approach against our method. The reviewer mentions concerns of potential unstable training with our approach, but we do not observe this in practice.
>
> **W2**) *Improvements potentially coming from better data (use of WebLi).*
> We would like to point out experimental results in the submitted version of the paper which already suggest that the gains are mainly coming from a better training method, rather than better data. Table 6 (in the appendix) ablates dataset versions according to our curation pipeline, showing that our curated dataset leads to moderate gains in NYUv2 (from 0.698 to 0.62 RMSE), when using a standard CLIP method. However, Table 1 indicates a much larger gain by changing from CLIP to our method, from 0.62 to 0.478 RMSE, which is roughly 2X the gain from data curation.
> Additionally, we are working hard on providing experimental results with a public dataset, as suggested by the reviewer. This is a significant engineering task, which consumes a very large amount of resources, not only for training, but also for downloading, curating and re-captioning the large-scale datasets. Given that WebLi and the other public datasets are collected in similar ways, we believe that other datasets of the same size will yield similar results. Nevertheless, we are doing our best to provide results on this as soon as possible.

---

> > ### Author Response · Authors · 2024-11-22
> >
> > We have now uploaded a new PDF version with the latest modifications to the text, as per suggestions from all reviewers. We are continuing to work on the additional experiments suggested by the reviewer and will report back once they are ready.
> >
> > Thanks once again for your attention. We continue to be available for discussions in case any further clarifications can be helpful.

---

> > > ### Author Response · Authors · 2024-11-26
> > >
> > > We have now uploaded a second revised PDF version with the latest modifications to the manuscript.
> > >
> > > We have included an experiment replacing WebLI by DataComp, as suggested by the reviewer (**W2** above): see the new Table 7, and associated paragraph “Training on DataComp”, in appendix A.1. Results show that very similar performance is obtained if using WebLI or DataComp. This strongly suggests that improvements over previous methods do **not** come from better data sources: for example, CLIP trained on WebLI or DataComp leads to very similar numbers (same observation for TIPS).
> > >
> > > We are continuing to work on the remaining experiment suggested by the reviewer (comparison to EVA’s training approach), and will report back once it is ready.
> > >
> > > Thanks once again for your attention. We continue to be available for discussions in case any further clarifications can be helpful.

---

> > > > ### Author Response · Authors · 2024-11-27
> > > >
> > > > We have now uploaded a final revised PDF version with the latest modifications to the manuscript.
> > > >
> > > > We have included an experiment ablating the way to combine contrastive and self-supervised learning, as suggested by the reviewer (**W1** above). The result can be found in Tab 8 (E). While the successive manner of CLIP → MIM training improves on dense tasks compared to the CLIP baseline, the proposed TIPS approach to combine contrastive learning with self-distillation and MIM simultaneously performs better across the board.
> > > >
> > > > We believe that all concerns from the reviewer have been addressed at this point, and we thank the reviewer for the attention here. We continue to be available for discussions in case any further clarifications can be helpful.

---

> > > > > ### Comment · Reviewer_6Wzj · 2024-11-29
> > > > >
> > > > > Thanks for the detailed reply and the new results. My concerns about the performance and the proprietary WebLI dataset have been addressed. I appreciate the new experiments conducted during the rebuttal, especially considering the large training cost of ablation on the training pipeline and the dataset choice. However, I still think the method is not that new and inspiring since the two core methods have already been separately validated in previous work. Overall, I think the results presented in the paper are valuable for the community. After reading other reviews, I would keep my initial rating and recommend acceptance for this paper.

---

> > > > > > ### Author Response · Authors · 2024-11-29
> > > > > >
> > > > > > Thanks for the reply. We are glad to see that the reviewer's concerns were addressed, and that the reviewer recognizes our results are valuable for the community, leading to an acceptance recommendation.

---

### Author Response · Authors · 2024-11-20
**Rebuttal comment to all reviewers**

We would like to thank all reviewers for the valuable feedback.

We are encouraged that our work is generally recognized as “**solid**” (6Wzj), exploring an “**important direction**” (BGwY). Reviewers acknowledged that our method is “**intuitive and promising**” (BGwY), “**novel**” (FXGp), showing “**strong generalizability**” (wggG), “**effective**” (FXGp), overall achieving “**significant**” results (wggG). The experiments are regarded as “**comprehensive and promising**” (BGwY), “**detailed**” (wggG), “**extensive**” (6Wzj), “**substantial**” (wggG), with ablations that are “**helpful**” (6Wzj). In terms of presentation, the paper is “**well written**” (BGwY), “**well-structured and clearly articulated**” (wggG).

We also truly appreciate the constructive comments, which help us improve our work and strengthen the paper. Given that there are no significant concerns that are common across all reviewers, we will address them directly in the individual sections below. We are doing our very best to address all of the comments and are committed to discussing with the reviewers in detail to help with the paper’s assessment.

Please note that we are currently working on the requested experiments and will provide their results as soon as they are ready. In any case, we wanted to start discussion with all reviewers as soon as possible, in order to provide ample time for discussions.

---

> ### Author Response · Authors · 2024-11-22
>
> Dear reviewers,
>
> We have now uploaded a new PDF version with the latest modifications to the text, as per reviewer suggestions. Please note that the changes are marked in blue, to make them more salient. The main changes were:
> - Ablation on SSL components, according to reviewer BGwY (Q1): Table 7 (*relabeled to Tab 8 in latest version*) was added to the appendix, with the associated paragraph “Ablation on self-supervised learning components”. These experiments report results varying the masking approach and ratio for the masked modeling component of TIPS. Additional ablations will be included in the next few days.
> - Formatting changes, according to reviewer FXGp (W1): we fixed the spacing issues around figures 1 and 2, as requested. We additionally improved formatting as follows: (i) rearranging the placement of some tables and fixing the numbering of Tables 3 and 4; (ii) improving spacing in the new Table 3 (compare to previous Table 4); (iii) enhancing formatting of Tab 2 to guide the reader more effectively over all results.
> - MaskCLIP discussion, according to reviewer BGwY (W2): included the MaskCLIP reference, which was previously missing. Discussed it in Related Work (section 2) and section 3.2, in relation to the proposed method. Added MaskCLIP results in Tab 3.
>
> Thanks once again for your attention. We continue to be available for discussions in case any further clarifications can be helpful.

---

> > ### Author Response · Authors · 2024-11-26
> >
> > Dear reviewers,
> >
> > We have now uploaded a second revised PDF version with the latest modifications to the manuscript, as per reviewer suggestions. Please note that all changes in the revised versions are marked in blue, to make them more salient. The main changes in this iteration were:
> > - Experiment replacing the WebLI training set by DataComp, as per request of reviewer 6Wzj (W2): see new Table 7 and associated paragraph “Training on DataComp”, in appendix A.1. This experiment shows that similar results are obtained if training on WebLI or DataComp, indicating the effectiveness of TIPS independently of the training dataset.
> > - Additional ablation on SSL components, as per request of reviewer BGwY (Q1): the new Table 8 now reports ablations considering additional image augmentations, and related discussion was added to the paragraph “Ablation on self-supervised learning components”.
> > - Dual embedding attention visualization, as per request of reviewer BGwY (Q2): new appendix section A.6 was added, with detailed visualizations and discussions on the roles of the two embeddings, which corroborate our intuitions that the two embedding heads focus on different aspects of the image.
> >
> > Thanks once again for your attention. We continue to be available for discussions in case any further clarifications can be helpful.

---

> > > ### Author Response · Authors · 2024-11-27
> > >
> > > Dear reviewers,
> > >
> > > We have now uploaded a final revised PDF version with the latest modifications to the manuscript, as per reviewer suggestions. Please note that all changes in the revised versions are marked in blue, to make them more salient.
> > >
> > > The final change in this iteration is the inclusion of the experiment comparing the EVA-like successive CLIP → MIM training, against our proposed learning process. This was requested by reviewer 6Wzj (W1). The new result can be found in Tab 8 (E), showing that our proposed method of combining contrastive and self-supervised learning simultaneously performs better.
> > >
> > > Thanks once again for your attention. We continue to be available for discussions in case any further clarifications can be helpful.

---

### Meta-Review · Area_Chair_Bp1Y · 2024-12-17

**Metareview:**

The paper presents an approach to integrating spatial awareness into text-image pretraining. The reviewers generally agree on the paper's strengths, including its extensive experiments, strong generalization across tasks, and intuitive dual-embedding technique. Some concerns were raised about the novelty of combining existing methods, though these were largely addressed during the rebuttal phase. Given the positive scores after the rebuttal and discussion period, the AC recommends acceptance.

**Additional Comments On Reviewer Discussion:**

Reviewers raised concerns about novelty, dataset dependency, and marginal improvements. The authors conducted new experiments comparing datasets, ablated self-supervised components, and visualized dual embeddings, addressing all issues. Synthetic captions’ role in spatial tasks was clarified, and formatting improved. While novelty was questioned, reviewers acknowledged the method’s effectiveness and comprehensive results. Given the thorough rebuttal, additional experiments, and general agreement on contributions, the AC weighed these responses positively, leading to the final decision to recommend acceptance.

---

### Decision · Program_Chairs · 2025-01-22

Accept (Poster)